

**Seasonal characteristics of emission, distribution and radiative effect of marine organic aerosols**
**over the western Pacific Ocean: an analysis combining observations with regional modeling**
Jiawei Li[1], Zhiwei Han[1,2*], Pingqing Fu[3], Xiaohong Yao[4]
[1]CAS Key Laboratory of Regional Climate-Environment for Temperate East Asia (RCE-TEA),
Institute of Atmospheric Physics, Chinese Academy of Sciences, Beijing 100029, China
[2]University of Chinese Academy of Sciences, Beijing 100049, China
[3]School of Earth System Science, Tianjin University, Tianjin 300072, China
[4]Laboratory of Marine Environmental Science and Ecology, Ministry of Education, Ocean University
of China, Qingdao 266100, China
**Abstract**: Organic aerosols from marine sources over the western Pacific Ocean of East Asia were
investigated by using an online-coupled regional chemistry-climate model RIEMS-Chem for the entire
year 2014. Model evaluation against a wide variety of observations from research cruises and in-situ
measurements demonstrated a good skill of the model in simulating temporal variation and spatial
distribution of particulate matter with aerodynamic diameter less than 2.5 μm and 10 μm ($PM_{2.5}$ and
$PM_{10}$), black carbon (BC), organic carbon (OC), and aerosol optical depth (AOD) in marine atmosphere.
The inclusion of marine organic aerosols apparently improved model performance on OC aerosol
concentration, reducing the normalized mean biases from -19% to -13% (KEXUE-1 cruise) and -21%
to -3% (Huaniao Island) over the marginal seas of east China, and from 33% to 5% (Dongfanghong II
cruise) and from -13% to 3% (Chichijima Island) over remote oceans of the western Pacific. It was
found that marine primary organic aerosol (MPOA) accounted for majority of marine organic aerosol
(MOA) mass in the western Pacific. High MPOA emission mainly occurred over the marginal seas of
China and remote oceans of the western Pacific northeast of Japan. The seasonality of MPOA emission
is determined by the combined effect of Chlorophyll-a (Chl-a) concentration and sea salt emission flux,
exhibiting the maximum in autumn and the minimum in summer in terms of domain average over the
western Pacific. The annual mean MPOA emission rate was estimated to be $0.16 \times 10^{-2}$ μg m$^{-2}$ s$^{-1}$,
yielding an annual MPOA emission of 0.78 Tg yr$^{-1}$ over the western Pacific, which potentially accounted
for approximately 8~12% of global annual MPOA emission. The regional and annual mean near surface


MOA concentration was estimated to be 0.27 μg m$^{-3}$ over the western Pacific, with the maximum in
spring and the minimum in winter, resulting from the combined effect of MPOA emission, dry and wet
depositions. Marine secondary organic aerosol (MSOA) produced by marine biogenic VOCs (isoprene
and monoterpene) was approximately 1~2 orders of magnitude lower than MPOA. The simulated annual
and regional mean MSOA was 2.2 ng m$^{-3}$, with the maximum daily mean value up to 28 ng m$^{-3}$ over the
western Pacific in summer. MSOA had a distinct summer maximum and winter minimum in the western
Pacific, generally consistent with the seasonality of marine isoprene emission flux. In terms of annual
mean, 26% of the total organic aerosol concentration was contributed by MOA over the western Pacific,
with an increasing importance of MOA from the marginal seas of China (13%) to remote oceans of the
western Pacific (42%). MOA induced a minor direct radiative effect (DRE), with a domain and annual
mean of -0.21 W m$^{-2}$ at the top of the atmosphere (TOA) under all-sky condition over the western Pacific,
whereas the mean indirect radiative effect (IRE) due to MOA at TOA (IRE$_{MOA}$) was estimated to be -
4.2 W m$^{-2}$. MSOA contributed approximately 6% of the annual and regional mean IRE$_{MOA}$ over the
western Pacific, with the maximum seasonal mean contribution up to 14% in summer, which meant
MPOA dominated the IRE$_{MOA}$. It was noteworthy that the IRE$_{MOA}$ accounted for approximately 32% of
that due to all aerosols over the western Pacific of East Asia, indicating an important role of MOA in
perturbing cloud properties and shortwave radiation in this region.
**1 Introduction**

Atmospheric aerosol is one of the most important and uncertain factors in climate change issues

(IPCC, 2013). Aerosols can alter radiation balance by scattering/absorbing solar/infrared radiation, and
affect cloud microphysics and lifetime by activating as cloud condensation nuclei (CCN), exerting
significant effects on climate system directly and indirectly. Aerosols are originated from anthropogenic
and natural sources and of high spatial and temporal variability and short atmospheric lifetime relative
to greenhouse gases. Consequently, aerosol radiative and climatic effects often have strong regional
characteristics.

The western Pacific Ocean is frequently influenced by continental outflow of both anthropogenic

and natural aerosols. Due to continuous growth of economy and energy consumption in the past decades,
the aerosol level in China has been enhanced (Smith et al., 2011; Li et al., 2017) and may have
potentially significant effects on radiation and cloud over not only the East Asian continent but also the



wide downwind oceanic areas. Besides, East Asia is one of the major dust source regions on earth (Shao
and Dong, 2006). Dust storms often occur in spring and dust particles can be transported eastward from
the deserts and Gobi areas of north China and southern Mongolia to the western Pacific Ocean (Gong
et al., 2003), providing nutrients (e.g. iron) for phytoplankton or even triggering the outbreak of algae
bloom in oceans (Calil et al., 2011; Tan et al., 2017). In addition to anthropogenic and dust aerosols,
marine aerosols also significantly affect aerosol chemical composition, radiation transfer, and cloud
properties in marine atmosphere. The behaviors and climatic impacts of sea salt and non-sea-salt sulfate
oxidized from dimethylsulphide (DMS) have been extensively investigated (Graf et al., 1997; Liao et
al., 2004; Rap et al., 2013). In recent years, particular attentions have been paid on the sources and
impacts of marine organic aerosols (O'Dowd et al., 2004; Meskhidze and Nenes, 2006; Luo and Yu,
2010; Vignati et al., 2010; Gantt et al., 2011; Huang et al., 2018), however, such studies were still very
limited, especially for the western Pacific.

O'Dowd et al. (2004) found that organic matter dominated the chemical composition of marine

aerosol during plankton bloom periods from spring to autumn over the North Atlantic Ocean,
contributing 63% to sub-micron aerosol mass. Meshkidze and Nenes (2006) revealed a significant
impact of phytoplankton bloom on cloud droplet number concentration and radiation balance in the
Southern Ocean and proposed a major contribution of secondary organic aerosol (SOA) from
phytoplankton produced isoprene. Some studies indicated that primary marine sources may dominate
marine organic matter, whereas SOA oxidized from marine isoprene could only comprise a small
fraction of the observed organic aerosol mass over marine environment (Facchini et al. 2008; Arnold et
al., 2009; Myriokefalitakis et al., 2010). The estimated global emission amounts of primary marine
organic matter varied largely among models. Using the global aerosol-climate model ECHAM5-HAM,
Roelofs (2008) estimated a global production of marine organic aerosols to be 75 TgC yr$^{-1}$. Spracklen
et al. (2008) estimated the marine organic carbon emission to be approximately 8 TgC yr$^{-1}$ based on
measured organic carbon mass and satellite retrieved chlorophyll-a (Chl-a) concentration. Vignati et al.
(2010) derived a global emission of marine primary organic matter in the sub-micron size by sea spay
process to be 5.8 TgC yr$^{-1}$ by using an off-line global Chemistry-Transport Model TM5 with a
parameterization relating organic emission fraction to sea surface Chl-a concentration. Gantt et al. (2011)
found that the combination of 10 m wind speed and sea surface Chl-a concentration were the most
consistent predictors of organic mass fraction of sea spray aerosol based on observations from the Mace



Head atmospheric research station on the Atlantic coast of Ireland and a site at the Point Reyes National
Seashore on the Pacific coast of California. They developed a new MPOA emission function and
estimated the global annual MPOA emission associated with sea spray to be from 15.9 TgC yr$^{-1}$ to 18.7
TgC yr$^{-1}$ (2.8~5.6 TgC yr$^{-1}$ in the sub-micron size). Regarding the influence on climatic factors, such as
cloud condensation nuclei (CCN), Ovadnevaite et al. (2011) revealed that MPOA was a dichotomy of
low hygroscopicity and high CCN activity through analysis of ambient measurements of aerosol
chemical compositions and size distributions at the Mace Head atmospheric research station, and
highlighted the importance of MPOA in CCN activation over marine atmosphere. A later study of
Westervelt et al. (2012) indicated that marine organic aerosols was able to increase CCN by up to 50%
in the Southern Ocean and by 3.7% globally during the austral summer based on the model simulation
of GISS GCM II'.

The above studies reveal the important role of marine organic aerosols in chemical composition,

radiation budget, and cloud microphysics with focus on the global scale. However, there is very limited
modeling research on this important and challenging issue for the western Pacific Ocean of East Asia.
To our knowledge, only two of our previous studies explored the effects of MPOA on chemical
composition, radiation and cloud over the western Pacific in springtime with an online-coupled regional
chemistry/aerosol-climate model RIEMS-Chem (Han et al., 2019; Li et al., 2019), whereas the
seasonality and annual aspect of MPOA and MSOA produced by marine isoprene and terpene are still
unknown. In this study, we conducted a one-year simulation with the developed RIEMS-Chem to further
explore the characteristics and radiative impacts of marine organic aerosols over the western Pacific.
The model simulated aerosol compositions were validated against a series of observations from ground
and cruise measurements, and the simulated MSOA was evaluated by comparison with cruise measured
secondary organic tracer in marine air masses. To our knowledge, for the first time, the seasonality of
emissions, concentrations, direct and indirect radiative effects of marine organic aerosols was
characterized and the annual means were estimated specifically for the western Pacific and for the key
oceanic regions of concern over East Asia. This study would provide new insights into properties and
impacts of marine organic aerosols over the western Pacific and would be a necessary supplement to
the global perspective of marine organic aerosols.

2 Model and data



### 2.1 Model description

An online-coupled regional atmospheric chemistry/aerosol-climate model RIEMS-Chem was used to investigate marine organic aerosols in this study. RIEMS-Chem composes of the host regional climate model RIEMS (Fu et al., 2005; Xiong et al., 2009; Wang S.Y. et al., 2015) and a comprehensive atmospheric chemistry/aerosol module. RIEMS was developed based on the dynamic structure of the fifth-generation Pennsylvania State University NCAR Mesoscale Model (MM5; Grell et al., 1995) with a series of parameterizations to represent major physical processes, such as a modified Biosphere-Atmosphere Transfer Scheme (BATS; Dickinson et al., 1993) for land-surface process, the Medium-Range Forecasts scheme (MRF; Hong and Pan, 1996) for planetary boundary layer process, the Grell cumulus convective parameterization scheme (Grell, 1993) for convective process, the Reisner explicit moisture scheme (Reisner et al., 1998) and a modified radiation package of the NCAR Community Climate Model (CCM3; Kiehl et al., 1996) for radiation transfer processes with aerosol effect. RIEMS has participated in the Regional Climate Model Intercomparison Project (RMIP) for Asia and it was one of the best models in predicting surface air temperature and precipitation over East Asia (Fu et al., 2005).

Atmospheric chemistry/aerosol modules have been incorporated into RIEMS in recent years, establishing the online-coupled model RIEMS-Chem, which can account for the interactions among chemistry, radiation, cloud, and meteorology (Han, 2010; Han et al., 2012). The model includes atmospheric chemistry and aerosol processes, such as gas and aqueous phase chemistries which are represented by the CB-IV mechanism (Gery et al. 1989) and RADM scheme (Chang et al., 1987), respectively; thermodynamic equilibrium process is represented by the ISORROPIA II model (Fountoukis and Nenes, 2007); heterogeneous reactions between gaseous precursors and aerosols are also taken into account (Li and Han, 2010; Li J. W. et al., 2018).

Dry deposition velocity is represented by a size-dependent parameterization over different underlying surfaces (Han et al., 2004). Dry deposition velocity of particle is expressed as the inverse of the sum of resistant plus a gravitational settling term. Over sea or ocean surfaces, the quasi-laminar boundary layer (QBL) is supposed to be disrupted by bursting bubbles, resulting in an increase in downward movement of particles. The approach of Van den Berg et al. (2000) is used in which quasi-laminar resistance $r_b$ is determined by Brownian diffusion and impaction when QBL is intact, and by turbulence and washout velocity of particles by spray drops when QBL is broken down.

Below-cloud scavenging (BCS) of particles between cloud base and ground surface represents




capture processes of particle by falling hydrometeor through Brownian and turbulent shear diffusion,
interception and inertial impaction, and is parameterized by a scavenging rate, which is a function of
precipitation rate and collision efficiency of particle by hydrometeor (Slinn, 1984).
Totally 10 aerosol types are simulated in RIEMS-Chem, which are sulfate ($SO_4^{2-}$), nitrate ($NO_3^-$),
ammonium ($NH_4^+$), black carbon (BC), primary organic aerosol (POA), secondary organic aerosol
(SOA), anthropogenic primary PMs ($PM_{2.5}$ and $PM_{10}$), dust, and sea salt. Sulfate is mainly produced
from the oxidation of $SO_2$ by OH radical in gas phase and the oxidation of dissolved $SO_2$ by $H_2O_2$, $O_3$,
and metal catalysis in aqueous phase (Chang et al., 1987). Nitrate and ammonium are produced through
thermodynamic processes represented by the ISORROPIA II model (Fountoukis and Nenes, 2007). BC,
POA, and anthropogenic primary PMs are considered chemically inert. SOA formation from
anthropogenic and biogenic VOC precursors is treated by a bulk yield scheme from Lack et al. (2004),
with SOA yield of 424 μg m$^{-3}$ ppm$^{-1}$ for toluene, 342 μg m$^{-3}$ ppm$^{-1}$ for xylene, and 762 μg m$^{-3}$ ppm$^{-1}$ for
monoterpene. For irreversible conversion of marine VOCs to SOA, a 28.6% mass yield is assumed for
isoprene (Surratt et al., 2010, Meskhidze et al., 2011) and 30% for monoterpene (Lee et al., 2006).
Based on the observational analysis of aerosol mixing state in eastern China (Wu et al., 2017), an
internal mixing assumption is adopted for anthropogenic aerosols and they are externally mixed with
natural aerosols and the geometric mean radius and standard deviation of the internal mixture are
estimated to be 0.11 μm and 1.65, respectively. Natural aerosols (mineral dust and sea salt) are
represented by 5 size bins (0.1~1.0, 1.0~2.0, 2.0~4.0, 4.0~8.0 and 8.0~20.0 μm). The deflation of
mineral dust is represented by the scheme of Han et al. (2004). The generation of sea salt aerosol through
bubbles is based on the scheme of Monahan et al. (1986) and is modified by considering the influences
of sea surface temperature (SST) (Jaeglé et al., 2011) and relative humidity (RH) (Zhang et al., 2005).
The hygroscopic growth of aerosol is parameterized by a κ parameterization (Petters and
Kreidenweis, 2007). The hygroscopicity parameters (κ) for inorganic aerosol components, BC, POA,
SOA, dust, and sea salt are set to be 0.65, 0, 0.1, 0.2, 0.01 and 0.98, respectively (Riemer et al., 2010;
Liu et al., 2010; Westervelt et al., 2012). The aerosol refractive index and hygroscopicity (κ) of the
internally mixed aerosol are calculated by volume-weighting of the parameters for each aerosol
component. Aerosol optical parameters including extinction coefficient, single scattering albedo, and
asymmetry factor are calculated by a Mie-theory based method developed by Ghan and Zaveri (2007),
which is much faster than traditional Mie code with a similar level of accuracy and has been successfully





used in estimating aerosol optical properties over East Asia (Han et al. 2011).
RIEMS-Chem has been successfully applied in previous modeling studies on anthropogenic
aerosols, mineral dust and marine aerosols regarding spatial-temporal distributions, physical and
chemical evolutions, and radiative and climatic effects over East Asia (Han et al., 2012; 2013; 2019; Li
et al., 2014; 2016a; 2016b; 2019). It is now participating in the international model comparison project
MICS-Asia III (Model Inter Comparison Study for Asia phase III) and shows a good ability in predicting
aerosol concentrations and AOD over East Asia (Gao et al., 2018).

2.2 Anthropogenic, biomass burning, and biogenic emission inventories
Monthly mean anthropogenic emissions of sulfur dioxide ($SO_2$), nitrogen ($NO_x$), ammonia ($NH_3$),
non-methane volatile organic compounds (NMVOC), carbon monoxide (CO), BC, POA, and other
anthropogenic primary $PM_{2.5}$ and $PM_{10}$ in China for the year 2014 are obtained from the MEIC
inventory (Multi-resolution Emission Inventory for China) which was developed by Tsinghua
University (http://meicmodel.org, last access: 2020/01/20). Anthropogenic emissions outside China are
taken from the MIX inventory which was developed to support the Model Inter-Comparison Study for
Asia phase III (MICS-Asia III) and the Hemispheric Transport of Air Pollution (HTAP) projects (Li et
al., 2017). Both inventories of MEIC and MIX have the same resolution of 0.5 degree. Open biomass
burning emissions of aerosols and gas precursors for the year 2014 with a spatial resolution of 0.5 degree
are derived from the Global Fire Emissions Database, Version 4.0 (GFED4) on a daily basis (Giglio et
al., 2013). Monthly mean terrestrial biogenic emissions of isoprene and monoterpene with a spatial
resolution of 0.5 degree are derived from the Global Emissions Inventory Activity (GEIA,
http://www.geiacenter.org/, last access: 2020/01/20). All the above emission data are bilinearly
interpolated to the lambert projection of RIEMS-Chem.

2.3 Marine primary emissions
2.3.1 Primary organic aerosol
The size-resolved marine primary organic aerosol (MPOA) emission is parameterized based on the
method of Gantt et al. (2011; 2012a). A briefly introduction is provided below.
The emission rate of MPOA is the product of sea salt emission rate ($E_{ss}$) and organic matter fraction
of sea salt ($OM_{ss}$), i.e. $E_{MPOA}= \alpha \times E_{ss} \times OM_{ss}$. $\alpha$ is a tuning factor. $E_{ss}$ is simulated on every model time



step. OM$_{ss}$ is the unitless organic mass fraction of sea salt aerosol in the range of 0 – 1. It is expressed
as a function of wind speed, surface seawater Chl-a concentration, and aerosol size:
$$OM_{SS} = \frac{\left(\frac{1}{1+\exp(X(-2.63[Chl\text{-}a])+X(0.18U_{10}))}\right)}{1+0.03\exp(6.81D_p)} + \frac{0.03}{1+\exp(X(-2.63[Chl\text{-}a])+X(0.18U_{10}))} \quad (1),$$

where U$_{10}$ is wind speed at 10 meter (m s$^{-1}$) simulated online by RIEMS-Chem, D$_p$ is the diameter of
sea salt aerosol, and Chl-a is the surface seawater chlorophyll-a concentration (mg m$^{-3}$). The Level-3
daily mean Chl-a concentration retrievals with 9 km resolution from the VIIRS (Visible infrared
Imaging Radiometer) sensor onboard the Suomi National Polar-orbiting Partnership (SNPP) satellite
platform (OBPG, 2018) are obtained for model inputs and it can reflect day-to-day variation of sea
surface Chl-a concentration associated with phytoplankton bloom in the western Pacific. X is a unitless
adjustable coefficient and is set to 3 based on Gantt et al. (2012a). An OM/OC ratio of 1.4 was applied
to convert organic matter (OM) to OC. For the tuning factor α, Gantt et al. (2012a) suggested a factor
of 6 was able to minimize the relative model biases for the global model GEOS-Chem at two oceanic
sites (Mace Head in North Atlantic and Amsterdam Island in remote south Indian Ocean). In this study,
we found that a factor of 2 was optimal to obtain the least bias between model simulation and
observation over the western Pacific. The large difference in the choice of α suggests that the emission
rate of MPOA could be very regionally dependent. Because there was limited information in optical
properties of MPOA, the refractive index of anthropogenic POA was used instead.

2.3.2 Isoprene and monoterpene
Marine isoprene emission released by phytoplankton activities is parameterized in RIEMS-Chem
using the scheme of Gantt et al. (2009) which considers light sensitivity of phytoplankton isoprene
production and dynamic euphotic depth. The scheme is expressed as:
$$SW_{isop} = H_{max} \times [Chl\text{-}a] \times \int_0^{H_{max}} EF \ln(I)^2 dh \qquad (2),$$

where SW$_{isop}$ is surface seawater isoprene concentration (μg m$^{-3}$), EF is the emission factor of isoprene
released by phytoplankton, I is the ambient photosynthetically active radiation (PAR in the unit of μEm$^{-}$
$^2$ s$^{-1}$), H$_{max}$ is the total water depth which isoprene production can occur from the surface to the point
and calculated as:






$$H_{max} = -\ln(\frac{2.5}{I_0})\frac{1}{k_{490}}$$

(3),

where $I_0$ is the all-sky surface incoming solar radiation (W m$^{-2}$) provided by the model during simulation.
$I_0$ and I have an approximate relationship of 1 W m$^{-2}$ ≈ 2 μEm$^{-2}$ s$^{-1}$. The diffuse attenuation coefficient
values at 490 nm $k_{490}$ (m$^{-1}$) is also obtained from VIIRS satellite. The isoprene production is assumed
to occur when the light level is greater than 2.5 W m$^{-2}$ in surface sea water.

The sea-air flux of marine isoprene ($E_{isop}$ in the unit of μg m$^{-2}$ s$^{-1}$) is parameterized following the

method of Palmer and Shaw (2005), which can be expressed as $E_{isop}$=k×$SW_{isop}$, where k is the sea-air
exchange coefficient (cm h$^{-1}$), it is calculated as k=0.31×$U_{10}$×(660/Sc)$^{1/2}$, where Sc is the Schmit number
of Isoprene. Marine emission of monoterpene is scaled by 0.2 to those of isoprene following the
suggestion from Myriokefalitakis et al. (2010).

2.4 Aerosol activation

A physically based scheme (namely A-G scheme) developed based on classical Köhler theory by

Abdul-Razzak and Ghan (1998, 2000) is incorporated into RIEMS-Chem to represent aerosol activation
of cloud droplet. This scheme calculates cloud droplet number concentration ($N_c$) with not only aerosol
mass/number concentration, but also aerosol size distribution and composition, updraft velocity and
ambient supersaturation. A-G scheme is computationally efficient with prediction accuracy of activation
fraction within 10% of that from detailed numerical model under a variety of atmospheric conditions.

Aerosols are activated if their critical supersaturation is less than the maximum ambient

supersaturation. The critical supersaturation for activating particles is determined by curvature effect
and solute effect. There is little information for physical properties of marine organic aerosols, some
key parameters, i.e. the number of ions the salt dissociates into water, the osmotic coefficient, the mass
fraction of soluble material, the density, and molecular weight are set to 3.0, 1, 0.1, 1.5 g m$^{-3}$, and 100,
respectively, according to a few previous studies (Abdul-Razzak and Ghan, 2004; Roelofs, 2008). The
soluble mass fraction of MSOA is assumed to be 0.2, slightly higher than that of MPOA. The size
distribution of marine organic aerosols is critical to aerosol activation and it is derived from cruise
measurements from Feng et al. (2017) over the western Pacific during the same period as this study, in
which the geometric mean diameter of marine organic aerosol number concentration (majority of which
is MPOA) was estimated to be approximately 0.1 μm, with the standard deviation of 1.6. MPOA can be





267 mixed with sea salt both externally or internally, and it is more likely to be externally mixed with sea

268 salt for finer aerosols (<200 nm in diameter) (Gantt and Meskhidze, 2013) and the effect of externally

269 mixed MPOA was found to be much more important than that of internally mixed MPOA (Gantt et al.,

270 2012b), so an external mixture of MPOA and sea salt is assumed in this study, which means additional

271 marine organic aerosols are produced to affect cloud properties and represents an upper limit of indirect

272 effect. The maximum ambient supersaturation is calculated by solving supersaturation balance equation

273 (Abdul-Razzak and Ghan, 1998). The updraft velocity is represented by the sum of grid mean updraft

274 velocity and subgrid updraft velocity, which is diagnosed from vertical eddy diffusivity according to

275 Ghan et al. (1997). The A-G scheme in RIEMS-Chem has been applied over the western Pacific Ocean

276 in spring 2014 and its prediction for hourly CCN concentration at different supersaturations has been

277 validated by cruise measurements from the marginal seas of China to remote oceans southeast of Japan,

278 which demonstrates a good ability, with the correlation coefficient of 0.87 and normalized mean bias

279 within 20%. More details on the treatment and evaluation of marine aerosol activation refer to Han et

280 al. (2019).

281  Once $N_c$ is derived from the above scheme, the cloud droplet effective radius $r_e$ is calculated

282 following the method of Martin et al. (1994). The number of aerosols activated is assumed to be equal

283 to the number of aerosols scavenged in cloud. The autoconversion rate from cloud water to rainwater

284 (second indirect effect) is considered and parameterized by the scheme of Beheng (1994), which

285 depends on $N_c$ associated with aerosols and cloud liquid water content. The effect of aerosols on ice

286 nuclei and convective cloud is not treated in this model due to limited knowledge at present.

288 2.5 Model setup and experiment design

289  This study focused on the western Pacific Ocean of East Asia. The model domain covered most

290 areas of eastern China, the Korean Peninsula, Japan, parts of Southeast Asia, and a wide area of the

291 western Pacific Ocean (Figure 1). A lambert conformal projection with 60 km horizontal resolution was

292 applied in the model. 16 vertical layers stretched unevenly from the surface to tropopause in a terrain-

293 following sigma coordinate with the first 8 layers within planetary boundary layer. The simulation

294 period was from 1 December 2013 to 31 December 2014 with the first month as model spin-up and the

295 whole year of 2014 was used for analysis. Final reanalysis data with 1°×1° resolution and 6-hour interval

296 from the National Centers for Environmental Prediction (NOAA/NCEP, 2000) was used to provide



initial and boundary conditions for meteorology. Chemical results derived from the MOZART-4 (Model
for Ozone and Related chemical Tracers, version 4; Emmons et al., 2010) simulation with 6-hour
interval were used to provide lateral conditions for trace gases and aerosols. Two simulations were
conducted. The full simulation (FULL) considered all anthropogenic and natural emissions, while the
NoMOE simulation shuts down all marine organic emissions (including MPOA, isoprene, and terpene).
The impacts of marine organic aerosols can be derived from the difference between the FULL and
NoMOE simulations (FULL minus NoMOE).

3 Model validations

In this section, the model results for OC, BC, $PM_{10}$, and $PM_{2.5}$ concentrations were compared with

a variety of observations from cruise, islands, and monitoring networks to help evaluate the model
ability over wide areas from eastern China to the western Pacific Ocean. Because the above comparison
was for total OC mass concentration, we also compared the simulated SOA from marine sources to
cruise measured SOA tracer to examine the model performance for marine organic aerosols.

3.1 Particulate matters ($PM_{10}$ and $PM_{2.5}$) and gas precursors

In-situ measurements of $PM_{10}$, $PM_{2.5}$, and gas precursors ($O_3$, $SO_2$, and $NO_x/NO_2$) at coastal and

island sites in Japan and Republic of Korea were obtained from EANET (Acid Deposition Monitoring
Network in East Asia, http://www.eanet.asia, last access: 2020/01/23) (Figure 1). Hourly concentrations
of $PM_{10}$, $SO_2$, $NO_x$ in Japan, $NO_2$ in Korea, and $O_3$ were automatically monitored at six Japanese sites
(Rishiri, Tappi, Sado-seki, Oki, Hedo, and Ogasawara) and three Korean sites (Jeju, Kanghwa, and
Imsil), whereas hourly $PM_{2.5}$ concentrations were only available at three Japanese sites (Rishiri, Sado-
seki, and Oki). Observations of hourly $PM_{10}$ and $PM_{2.5}$ concentrations in three major coastal cities of
China (Qingdao, Shanghai, and Fuzhou) were also collected from the CNEMC (China National
Environmental Monitoring Center, http://www.cnemc.cn/, last access: 2020/01/23) and used for model
comparison (Figure 1). As particulate matter in remote marine atmosphere is mainly composed of sea
salt, the model performance for $PM_{10}$ and $PM_{2.5}$ may reflect the model ability for sea salt simulation,
which is crucial to the estimation of MPOA emission.

Because the focus of this study is seasonal variation, the hourly $PM_{10}$ and $PM_{2.5}$ observations and

corresponding simulations were averaged to be monthly means and shown in Figure 2. In general,





RIEMS-Chem performed quite well in simulating monthly variation of $PM_{10}$ concentrations    at both
the EANET sites (Figure 2a~2i) and CNEMC sites (Figure 2j~2l) for the year 2014, although model
biases still occurred at some sites, such as the underprediction in winter and spring in Jeju (Figure 2g)
and Imsil (Figure 2i) and the overprediction in May in Oki (Figure 2d) and Rishiri (Figure 2a). It was
striking that $PM_{10}$ concentration peaked in May and was lowest in August at all Korean sites and
northern Japanese sites over northeast Asia (Figure 2a~2d and 2g~2i), which could be attributed to the
long-range transport of mineral dust from north China and Mongolia in spring and to the southwesterlies
consisting of mainly marine air masses in summer. It was noteworthy that the model simulated
seasonality and magnitude of $PM_{10}$ agreed quite well with observations at the four island sites of
northern Japan (Rishiri, Tappi, Sado, and Oki) (Figure 2a~2d), where sea salt aerosol played a more
important role than those sites in Korea, implying sea salt concentrations could also be well reproduced
by the model. The seasonality of $PM_{10}$ concentration at Hedo (Figure 2e) was different from above,
showing high values in winter as well besides the peaks in spring, which indicated potential influence
of continental anthropogenic sources under prevailing northwesterlies. The $PM_{10}$ level at Ogasawara
(Figure 2f) was much lower than those at the other sites and its seasonality was characterized by the
minimum in summer (5 $\mu g\ m^{-3}$) and the maximum in spring. The model reasonably reproduced the
seasonality at Hedo (Figure 2e) and Ogasawara (Figure 2f) as well, although it generally predicted lower
values at Hedo and higher values at Ogasawara. As for $PM_{10}$ concentrations at the CNEMC sites of
eastern China, the model simulated $PM_{10}$ concentrations very well for Shanghai (Figure 2k) and Fuzhou
(Figure 2l) in terms of both monthly variation and magnitude, showing higher values in spring and the
maximum in winter in Shanghai, and an almost stable level around 60 $\mu g\ m^{-3}$ in Fuzhou throughout the
year except for the elevated value in January. The $PM_{10}$ level in Qingdao (Figure 2j) was higher than
those in Shanghai and Fuzhou, and reached the maximum of 170 $\mu g\ m^{-3}$ in January due to anthropogenic
sources and the peak in March was resulted from the effect of mineral dust.

The monthly variations of $PM_{2.5}$ concentrations at Rishiri, Sado, and Oki (Figure 2m~2o) were

similar to those of $PM_{10}$, but the peaks in May were not as evident as those of $PM_{10}$, because mineral
dust comprises a small fraction of fine particles and has less effect on $PM_{2.5}$ variation. The model
reproduced $PM_{2.5}$ concentrations very well at the three coastal sites of eastern China (Figure 2p~2r) and
the monthly variation of $PM_{2.5}$ concentrations resembled those of $PM_{10}$, because fine particle accounts
for a large fraction of PM mass in these Chinese megacities due to the dominant effect of anthropogenic



sources.

Table 1 shows that for all the 9 EANET sites, the overall mean $PM_{10}$ concentration was 30.0 μg m$^-$

$^3$ from observation and 28.5 μg m$^{-3}$ from simulation, with the overall correlation coefficient (R) of 0.65
(0.48~0.64) and the normalized mean bias (NMB) of -5% (-27~36%). For $PM_{2.5}$, the mean
concentrations averaged over the EANET sites were 10.9 μg m$^{-3}$ from observation and 12.3 μg m$^{-3}$ from
simulation, with R and NMB of 0.61 (0.53~0.64) and 12% (0~21%), respectively. The annual mean
observed and simulated $PM_{10}$ concentrations at the 3 CNEMC sites (Table 2) were 81.6 μg m$^{-3}$ and 80.7
μg m$^{-3}$, with R and NMBs of 0.65 (0.38~0.61) and -1% (-4~1%), respectively, while the annual mean
observed and simulated $PM_{2.5}$ concentrations, R, and NMB were 46.6 μg m$^{-3}$, 43.4 μg m$^{-3}$, 0.70
(0.44~0.72), and -7% (-12~0%), respectively. The good performance statistics shown in Table 1 and
Table 2 suggest a good skill of RIEMS-Chem in reproducing PM levels from the coastal regions of east
China to the remote western Pacific. Figure 2, Table 1, and Table 2 also illustrate that the spatial
distribution of PM exhibited higher concentrations at the continental (costal) sites (CNEMC sites, Jeju,
Kanghwa, and Imsil) and lower concentrations at the remote island site (Ogasawara) over the western
Pacific, which were also reasonably reproduced by RIEMS-Chem.

Seasonal mean statistics of $PM_{10}$ and $PM_{2.5}$ concentrations at the EANET and CNEMC sites were

also listed in Table 1 and Table 2. Statistics for spring (March-April-May, MAM), summer (June-July-
August, JJA), autumn (September-October-November, SON), and winter (December-January-February,
DJF) were calculated. $PM_{10}$ observations generally exhibited higher concentrations in MAM and DJF,
moderate concentrations in SON, and lower concentrations in JJA at most sites covering coastal areas
(CNEMC sites, Jeju, Kanghwa, and Imsil) and remote islands (e.g. Oki, Hedo, and Ogasawara). The
model reproduced such seasonal variation of $PM_{10}$ reasonably well although some underestimations
occurred from winter to spring at Jeju and Imsil (Figure 2g, 2i), which could be attributed to the
uncertainties in emissions (anthropogenic, biomass burning).

In all, RIEMS-Chem was able to reproduce the spatial distribution and seasonal variation of $PM_{10}$

and $PM_{2.5}$ concentrations over the western Pacific. The good performances of $PM_{10}$ and $PM_{2.5}$ in the
marine environment of less anthropogenic source influence also imply that the model may be able to
reproduce sea salt reasonably well, which is essential to the estimation of marine MPOA emission.

In addition to the validation for PM concentrations, the model performances for gas precursors ($O_3$,

$SO_2$, and $NO_x/NO_2$) were also evaluated against hourly observations at the EANET sites (Table S1).



The model performance for $O_3$ concentration was generally well, with the overall R and NMB of 0.54
and 6%, respectively. The best performances for $O_3$ were at Hedo and Ogasawara, showing the R of
0.84 and a small NMB of 5~7%. The model fairly reproduced the variation and magnitude of $SO_2$
concentrations, with an overall R of 0.51 and an NMB of 10%. For $NO_x/NO_2$, the model performance
was not as good as those for $O_3$ and $SO_2$. On average, the $NO_x/NO_2$ concentration biased high by 36%
for all sites, with R of 0.48. Local emissions at some remote/rural sites which were unable to be captured
by the monthly mean emission inventory and the relatively coarse grid resolution could be partly
responsible for these biases. Despite the biases, the overall statistics were generally acceptable for gas
precursors, indicating that the atmospheric chemistry processes were reasonably represented by
RIEMS-Chem over the western Pacific.

3.2 Carbonaceous aerosols
Modeled BC and OC concentrations were compared with observations from research cruises and
from previous publications at coastal/remote islands. BC is considered to be inert and chemical inactive,
so it is governed solely by physical processes and a good indicator of long-range transport. The analysis
of BC can help assess the potential effect of marine organic emissions.

3.2.1 Measurements from research cruises
There were two research cruise campaigns covering the western Pacific during the spring and
summer of 2014 (Figure 1).
The spring cruise campaign was carried out from 17 March to 22 April 2014 onboard the research
vessel R/V Dongfanghong II, which started from Qingdao, sailed to the western Pacific Ocean, and then
returned (Figure 1) (Luo et al., 2016; Feng et al., 2017). OC and BC samples were collected by an 11-
stage MOUDI (Models110-IITM) (0.054~18 μm) equipped with pre-combusted quartz filters onboard
the vessel. Mass concentrations of total OC (primary and secondary) and BC were determined by the
thermal/optical carbon analyzer (Sunset Laboratory Inc., Forest Grove, OR). Totally 19 daily BC and
OC samples were collected during the cruise. Detailed information about this campaign and the
sampling and analysis techniques were documented in Feng et al (2017).
The early summer campaign was carried out from 18 May to 12 June 2014 (Kang et al., 2018).
Total suspended particles (TSP) were collected on pre-combusted quartz filters using a high-volume air



sampler (Kimoto, Japan) onboard the KEXUE-1 Research Vessel during a National Natural Science
Foundation of China (NSFC) sharing cruise (Figure 1). This campaign covered low- to mid-latitudes of
the western Pacific Ocean (over the Yellow Sea and the East China Sea). Totally 51 half-day
(daytime/nighttime) OC samples were obtained during this campaign. Detailed information about this
campaign and samples were described in Kang et al. (2018).

Figure 3a shows the observed and simulated daily BC concentrations along the cruise track during

the spring campaign. An obvious spatial gradient was found for BC concentration, which was
characterized by apparent higher concentrations of 0.5~4.2 µg m$^{-3}$ over the marginal seas of China (the
Yellow Sea and East China Sea, 18~19 March and 21~22 April) and very low concentrations of ~0 to
<0.2 µg m$^{-3}$ over open oceans (during most of the measurement days). It is interesting to note that an
observed BC peak occurred on 21 March, which could be attributed to the influence of biomass burning
sources over northeast Asia. This was demonstrated by backward trajectories and isotope analysis for
the same campaign (Luo et al., 2016; 2018), which showed that biomass burning aerosols were brought
from northeast Asian continent to the cruise positions on 19~21 March by continental outflows. The
model generally reproduced the spatial and temporal variations of BC concentration during the
campaign period; however, the BC peak on 21 March was missed by the model simulation. Uncertainties
in biomass burning emission could be responsible for such model bias. On average, the measured and
simulated BC concentrations during this campaign onboard the Dongfanghong II cruise were 0.49 µg
m$^{-3}$ and 0.55 µg m$^{-3}$, respectively, with the R and NMB of 0.87 and 13% (Table 3).

Figure 3b shows the daily mean OC concentrations from observation and model simulation for the

same cruise. In general, the observed OC exhibited a similar spatial distribution and temporal variation
to that of BC, with higher concentrations over the marginal seas and relatively lower concentrations
over open oceans. The model generally captured the spatial-temporal features along the cruise track.
Like BC, the observed OC concentrations were high on 21 and 25~26 March mainly due to the
continental outflow of biomass burning emissions from northeast Asia, and the model largely
underpredict the high OC observation in these days. It is noteworthy that two OC peaks appeared on 10
and 12 April when the ship was over the open ocean east of Japan (the ship location was around 33.5°N,
146.0°E on 10 April and around 35.9°N, 144.0°E on 12 April, approximately 400~500 km to the east of
Japan), whereas the elevation of BC concentration was not evident. Because BC and OC are often
originated from the same anthropogenic and biomass sources, the inconsistency in daily variation





between BC and OC in these areas implied a potential influence of marine sources rather than that from
anthropogenic and biomass burning emissions. Coincidentally, during these days (10 and 12 April), Chl-
a concentrations over the oceanic areas east of Japan (the region of 35°N to 43°N and 140.0°E to
150.0°E, north to the ship location) reached as high as 45 mg m$^{-3}$, as a comparison, the monthly mean
Chl-a concentration in April over the same region was in a range of 2 to 14 mg m$^{-3}$. The apparent higher
Chl-a concentration implied enhanced marine primary organic emissions during these days. In addition,
northerly winds prevailed over this region during the period, which likely blew marine organic carbon
(marine-OC) aerosols produced from the intense bloom regions to the south where the ship located,
leading to the elevation of OC concentrations. It was impressive that the model reasonably captured the
two peaks on 10 and 12 April when considering marine organic aerosols (marine-OC in Figure 3b). The
cruise campaign average OC concentration was 1.20 μg m$^{-3}$ from observation and 1.14 μg m$^{-3}$ from
simulation, with the R and NMB of 0.66 and -5%, respectively (Table 3). The inclusion of marine-OC
(including both primary and secondary OC) reduced the model bias from -33% to -5% along the cruise.
The average contribution of marine-OC to the total OC mass in the marine atmosphere was
approximately 29% along the cruise, with lower contributions of 11~27% over the marginal seas of
China (18~19 March and 21~22 April) and higher contributions of 32~74% over the open oceans (5~18
April) (Figure 3b), demonstrating an increasing importance of marine organic aerosols to total OC mass
from the marginal seas to remote open oceans.

Shown in Figure 3c is OC samples collected onboard the KEXUE-1 Research Vessel over the East

China Sea during the early summer campaign and the corresponding model results along the cruise track.
It was impressive that there were four OC peaks observed during the campaign, with three occurring
over the northern parts of the East China Sea (on 20 May, 26~29 May, and 1~5 June) and one over the
southern part of the East China Sea on 22 May. The model reproduced the OC variation quite well
during most of the cruise track, capturing the three OC peaks over the northern parts of the East China
Sea although low biases occurred for the first peak (over the area of 27.5°N to 30.0°N and 121.6°E to
121.9°E). The model missed the second OC peak on 22 May over the southern part of the East China
Sea (over the area of 22°N to 23°N and 121.5°E to 122.2°E). Kang et al. (2018) proposed that this peak
was seriously affected by biogenic and biomass burning emissions from Southeast Asia (Philippines)
because the OC concentrations from 21 to 25 May were characterized by high abundance of
sesquiterpene-derived SOA which was mainly originated from terrestrial photosynthetic vegetation (e.g.





trees and plants). Uncertainties in emission inventories, such as missing some biogenic sources (e.g.
fungal spores) could be partly responsible to the model biases. In addition, some regions of Southeast
Asia (e.g. Philippines) were not included in the study domain, instead, their influence on the study
domain was represented by chemical boundary conditions from MOZART simulation, so, the
uncertainties in chemical boundary conditions may also contributed to such biases. At the time of the
third (25°N to 26°N and 118.8°E to 121.7°E) and fourth (28°N to 28.7°N and 119.6°E to 122.7°E) OC
peaks, the ship was close to the shore and predominately affected by continental sources (such as
anthropogenic and biomass burning emissions), the model captured the peaks quite well in terms of
both temporal variation and magnitude. On average, the observed and simulated OC concentrations
from the KEXUE-1 cruise were 4.26 μg m$^{-3}$ and 3.68 μg m$^{-3}$, respectively, with R and NMB of 0.75 and
-13% (Table 3). The inclusion of marine-OC reduced the NMB from -19% to -13%. Along the cruise
track, marine-OC was estimated to account for 6% (1~60%) of the total OC mass on average, with lower
contribution over the seas close to the continent (1~9%) and higher contribution over the seas far from
the continent (7~60%). During the KEXUE-1 cruise campaign, the contribution of marine-OC to total
OC mass was obviously lower than that during the spring campaign conducted by the Dongfanghong
II, because this cruise over the marginal seas of China was more affected by continental outflow of
anthropogenic and biomass emissions compared with that mainly over the open oceans.

3.2.2 Measurements at island and coastal sites

In this section, long-term observations of OC and BC obtained from previous publications were

collected and compared with the model simulation. The four datasets were introduced briefly below.

From 2001 to 2012, carbonaceous aerosol samples (OC and BC) in TSP were continuously

collected on a weekly basis at Chichijima Island (the same place as Ogasawara in Figure 1), a remote
island located in the western North Pacific, by Boreddy et al. (2018). The reported monthly mean OC
and BC concentrations of the 12-year average were used to verify the model performance over remote
oceans.

Long-term (2009–2015) observations of BC concentrations were conducted at Fukue Island of

western Japan using a continuous soot-monitoring system (COSMOS) (Figure 1) by Kanaya et al.
(2016). The reported monthly mean BC concentrations for the year 2014 and the 7-year average were
used in this study.





Measurements of seasonal mean OC and BC concentrations in TSP at Huaniao Island (a pristine
island about 100 km southeast of Shanghai over the East China Sea, Figure 1) from October 2011 to
August 2012 (Wang et al., 2015) and at Okinawa island (the same place as Hedo in Figure 1) in the
western Pacific Ocean from October 2009 to October 2010 (Kunwar and Kawamura, 2014) were
collected and used for model validation in this study.

3.2.2.1 BC
At Huaniao Island (Figure 4a), BC concentration was observed to be highest in winter (DJF),
followed by that in spring (MAM) and autumn (SON), and lowest in summer (JJA). The model generally
reproduced the seasonality but predicted higher BC concentrations in JJA (Figure 4a). The model biases
could be partly attributed to the differences in emission and meteorological conditions in different years
between observation and simulation. The simulated annual mean BC concentration at Huniao Island
was 1.2 μg m$^{-3}$, generally consistent with the observed 1.1 μg m$^{-3}$ (Table 4).
At Okinawa Island (Figure 4b), which is located in the outflow region of East Asian continent, BC
concentration exhibited the maximum in DJF, followed by that in MAM, and lower concentrations in
JJA and SON. Kunwar and Kawamura (2014) indicated that during winter and spring, this site was
significantly influenced by the continental outflow of polluted air masses from East Asian continent,
resulting in an elevation of BC level; in summer, this site was dominated by maritime clean air masses,
while in autumn, it was affected by both oceanic and continental air masses. The model well reproduced
the seasonal variation of BC concentration at Okinawa. Although some low biases occurred in winter
and summer, the model results were still within the observation deviations (Figure 4b). In addition to
the differences in emission and meteorological conditions between observation and the simulation year,
local emissions in Okinawa, which were not represented by the monthly emission inventory, could also
be responsible to such biases. On average, the annual mean BC concentrations were 0.29 μg m$^{-3}$ from
simulation, somewhat lower than the observation of 0.38 μg m$^{-3}$ (Table 4).
Monthly mean observations provide more details on seasonal variation trend. At Fukue Island
(Figure 4c), the observed seasonality of BC in 2014 (the same time period as this study) exhibited the
highest level in January, the second peak in May, the lowest level in August, and the increase of BC in
autumn. The model generally reproduced the seasonal variation trend, in particular, well capturing the
peaks in January and May, and the minimum in August. It should be mentioned that the base year of the





MIX emission inventory for Japan used in this study was 2010, so potential uncertainties in the emission
inventory could partly contribute to the model bias. The monthly variation of the 7-year average during
2009-2015 was similar to that in 2014, but with lower BC levels in January and May and higher levels
in autumn months (Figure 4c). On average, the modeled annual mean BC concentration was 0.44 μg m$^{-3}$,
18% higher than the observations for the year 2014 (0.37 μg m$^{-3}$) and for the 2009-2015 average (0.37
μg m$^{-3}$) (Table 5). The correlation coefficient between the monthly mean simulation and observation
was 0.79 for 2014.

Figure 4d shows the monthly mean BC concentrations averaged from long-term observations

(2001-2012) at Chichijima Island, far from the East Asian continent. The monthly variation of BC
observation at Chichijima Island generally resembled that at Fukue Island, except that BC concentration
peaked in March. The model reproduced the BC seasonality at Chichijima quite well except those in
January and February, when the model results were apparently larger than observations, which could be
due to the larger emission amounts in the Eastern Asian countries in the emission inventories of MIX
(2010) and MEIC (2014) than those during 2001-2012. The annual mean BC concentration was 0.14
derived from the monthly mean observation and 0.16 from simulation (Table 5), with an NMB of 11%
and a correlation coefficient of 0.88 at this site.

Both observations and simulations above illustrate that over the western Pacific, BC exhibited

higher concentrations in winter and spring due to the prevailing westerly winds in these seasons bringing
polluted air masses to the oceans (Figure 9b and 9c). The lowest BC concentration occurred in summer
over oceanic areas mainly due to the dominance of the pristine maritime air masses from open oceans
(Figure 9d). In autumn, both continental and oceanic air masses affected the western Pacific (Figure 9e),
leading to a moderate BC level in this season.

The above comparison of in-situ BC concentrations also revealed that the annual mean BC

concentration was approximately 1.1 μg m$^{-3}$ at Huaniao Island, decreased to the level of about 0.4 μg
m$^{-3}$ at the midway of the long-rang transport (at Fukue and Okinawa), and further dropped to 0.14 μg
m$^{-3}$ over remote oceans (at Chichijima). The model reasonably reproduced such spatial gradient of BC
mass, indicating a good skill of RIEMS-Chem in representing the physical processes and long-rang
transport of carbonaceous aerosols over the western Pacific.

3.2.2.2 OC





OC observations are limited in the western Pacific Ocean. We collected observations at islands from
previous publications (Boreddy et al., 2018; Kunwar and Kawamura, 2014; Wang F. W. et al., 2015)
for model comparison. Figure 5 shows the model simulated and observed seasonal/monthly mean OC
concentrations at the three islands over the East China Sea and the remote ocean of the western Pacific.
It should be kept in mind that the observations are averages of different years. At Huaniao Island (Figure
5a), a distinct seasonality of OC observation was shown, with the highest OC concentration of 4.7 μg
m$^{-3}$ in DJF, followed by 3.7 μg m$^{-3}$ in MAM and 3.8 μg m$^{-3}$ in SON, and the minimum of 1.1 μg m$^{-3}$ in
JJA (Table 4). It was encouraging that RIEMS-Chem reproduced the OC seasonality at Huaniao Island
quite well (Figure 5a), despite the different years between simulation and observation, indicating the
seasonal cycling of OC was typical and there were small changes in emission and meteorology between
2014 and 2011-2012. The simulated OC was also divided into OC originated from continental sources
(land-OC) and marine sources (marine-OC) to quantify the relative contribution of these sources to total
OC mass. The simulated annual mean OC concentration was 3.2 μg m$^{-3}$, in which 2.6 μg m$^{-3}$ (81%) was
contributed by land-OC and 0.6 μg m$^{-3}$ (19%) by marine-OC (Table 4). The simulation was very close
to the observation of 3.3 μg m$^{-3}$ (Table 4). It was striking that the inclusion of marine-OC obviously
improved the model performance, reducing the NMB from -21% to -3%. It was noteworthy that marine-
OC exhibited the maximum value in MAM and the minimum value in JJA. The higher Chl-a
concentration over the East China Sea in MAM might be responsible for the maximum at Huaniao
Island (Figure 7h and Table 7), whereas the lowest sea salt emission flux could result in the minimum
in summer (Table 7). In terms of seasonal mean, marine-OC accounted for 12%, 22%, 19%, and 23%
of the total OC concentration in DJF, MAM, JJA, and SON, respectively, with an annual mean
contribution of 19% at Huaniao Island. The lowest relative contribution (12%) of marine-OC in winter
was attributed to the maximum anthropogenic OC emissions in eastern China in this season.

At Okinawa (Figure 5b), the observed total OC showed the maximum in MAM, followed by that

in JJA, and the lower ones in DJF and SON during October 2009-2010. It was noteworthy that the
seasonality of OC was different from that of BC at Okinawa (Figure 4b). Figures 5a and 5b also show
that the seasonal cycling of OC concentration at Okinawa (Figure 5b) differed a lot from that at Huaniao
Island (Figure 5a), which indicated the differences in OC behavior between remote island and coastal
island. Kunwar and Kawamura (2014) suggested that continental outflows of polluted air masses were
mainly responsible for the elevated OC levels in spring and winter at Okinawa, whereas for the high



OC concentration in summer, SOA produced by local biogenic VOC emissions could be an important
source with respect to the large contribution from SOA to total OC (~48%). The model generally
reproduced the seasonal variation of OC except that it predicted lower OC level in summer, which could
be due to the exclusion of local biogenic VOC emissions in Okinawa in the GEIA emission inventory.
Zhu et al. (2016) also reported the largest biogenic isoprene emissions from local tropical trees in
summer at Okinawa and suggested that the VOC flux from trees dominated SOA over that from
surrounding seas in summer. In addition, as Okinawa is a resort place, local anthropogenic emissions
which were not well represented in the MIX emission inventory may also contribute to the model-
observation deviation. In terms of annual average, the observed OC concentration was 1.8 µg m$^{-3}$, larger
than the simulations of 1.3 µg m$^{-3}$ from the FULL case including marine-OC and of 1.1 µg m$^{-3}$ from the
NoMOE case excluding marine organic emissions (Table 4). The inclusion of marine organic emissions
improved OC simulation at Okinawa, reducing the NMB from -39% to -28%. It was estimated that
marine-OC accounted for 18%, 17%, 10%, and 18% of total OC mass concentration at Okinawa in DJF,
MAM, JJA, and SON, respectively, with an annual mean contribution of 17%. The relatively smaller
contribution of marine-OC to the total OC mass at Okinawa than that at Huaniao Island (19%) could be
attributed to the higher Chl-a concentration and MPOA emission flux in the marginal seas of China than
those over remote western Pacific south of Japan (Figure 7), although Huaniao Island was closer to the
continent.
Long-term average (2001-2012) of monthly mean OC concentrations at Chichijima Island reported
by Boreddy et al. (2018) and the simulated monthly mean OC concentration in 2014 were shown in
Figure 5c. The observations show higher OC levels from January to March mainly due to continental
outflows. It was noticed that the simulated OC levels in April-May were apparently higher than
observations, which could be associated with different time periods between observation and simulation,
and with potentially stronger continental outflows and bloom in spring 2014 than those of ten-year
averages. OC observations were relatively lower in summer and autumn due to the dominance of high-
pressure system and pristine ocean air mass over the western Pacific (Figure 9d and 9e). The model
tended to predict lower OC level in summer and autumn (Figure 5c). Boreddy et al. (2018) indicated
that in summer and autumn, OC at Chichijima was often influenced by long-range transport of biomass
burning plumes from Southeast Asia, which was not well represented in the model (using chemical
boundary conditions from MOZART-4 instead) and led to low model bias. On average, the annual mean



OC concentration was 0.76 µg m$^{-3}$ from observation, 0.78 µg m$^{-3}$ from the FULL case, and 0.66 from
the NoMOE case (Table 5). The inclusion of marine organic emissions reduced the annual mean NMB
from -13% to 3% and enhanced the correlation coefficient from 0.56 to 0.6 at this site. The apparent
better simulation from the FULL case indicated the necessity of inclusion of marine organic emissions
for simulating OC over the remote oceans of the western Pacific. Both observation and model simulation
revealed higher seasonal mean OC concentrations in MAM (observed: 0.83 µg m$^{-3}$, simulated: 0.91 µg
m$^{-3}$) and DJF (observed: 0.90 µg m$^{-3}$, simulated: 1.2 µg m$^{-3}$) when the measurement site was frequently
influenced by continental outflows, whereas lower concentrations in JJA (observed: 0.65 µg m$^{-3}$,
simulated: 0.47 µg m$^{-3}$) and SON (observed: 0.66 µg m$^{-3}$, simulated: 0.57 µg m$^{-3}$) when clean maritime
air masses or biomass burning plumes from Southeast Asia (e.g. Philippine) influenced this region. The
highest marine-OC concentration was 0.19 µg m$^{-3}$ in MAM, followed by 0.16 µg m$^{-3}$ in DJF and 0.11
µg m$^{-3}$ in SON, and the lowest one of 0.05 µg m$^{-3}$ in JJA. However, the percentage contribution of
marine-OC to the total OC mass was estimated to be largest in SON (20%), followed by 18% in DJF,
16% in MAM, and lowest in JJA (10%), with an annual mean contribution of 16% (Table 5). The largest
contribution in SON was associated with the relatively lower total OC concentration as shown in Figure
5c. The relative contribution from marine-OC to total OC at Chichijima Island resembled that at
Okinawa in terms of annual and season averages.

The above comparison against a variety of OC observations demonstrated a generally good skill of

RIEMS-Chem in simulating OC over the western Pacific in terms of seasonal variation and magnitude.
The better model results from the FULL case indicated that including marine organic emissions
apparently improved OC simulation over the western Pacific Ocean.

3.2.3 SOA over the western Pacific

Recently, Guo et al. (2020) reported SOA observations in the marine atmosphere from the marginal

seas of east China to the northwest Pacific Ocean. The measurements were conducted on three research
cruises in the spring and early summer of 2014 and in the spring of 2017. Total suspended particulate
(TSP) samples were collected from 19 March to 21 April 2014 over the northwestern Pacific Ocean
(NWPO), from 30 April to 17 May 2014 over the Yellow and Bohai seas (YBS), and from 29 March to
4 May 2017 over the South China Sea (SCS). SOA concentration was derived by using a tracer-based
method. The measured SOA concentrations were 467 ng m$^{-3}$ over the YBS, 617 ng m$^{-3}$ over the SCS,



and 155 ng m$^{-3}$ over the NWPO, respectively. The model simulated period and regional mean SOA
concentrations were 664 ng m$^{-3}$ over the YBS, 466 ng m$^{-3}$ over the SCS, and 157 ng m$^{-3}$ over the NWPO,
which were generally consistent with the above observations, although the study periods are not exactly
the same. Guo et al. (2020) also presents the tracer-based estimations of isoprene and monoterpene
derived SOA in the air masses from ocean (assuming marine sources), which were 1.7 ng m$^{-3}$ and 0.3
ng m$^{-3}$, respectively, over the western Pacific to the southeast of Japan, whereas the modeled SOA
concentrations produced from marine isoprene and monoterpene emissions along the cruise track were
1.55 ng m$^{-3}$ and 0.28 ng m$^{-3}$, respectively, generally agreeing with the tracer-estimation. However, it
should be mentioned that there could be uncertainties in such comparison. First, the isoprene- and
monoterpene-derived SOA tracers in the air masses categorized as marine sources by Guo et al (2020)
might include SOA tracers from terrestrial isoprene and monoterpene under the prevailing northwesterly
winds in spring, which could bias the estimation high; second, the measured tracer could just comprise
a part of total SOA tracers, which might bias the estimation low. Despite these uncertainties, the cruise
measured SOA concentration derived from marine isoprene and monoterpene was approximately
several ng m$^{-3}$ over the western Pacific, and it can reach approximately 10 ng m$^{-3}$ even through dividing
by a mass fraction of tracer compound to yield the concentration of total SOA tracers. It was noteworthy
that both observation and model simulation exhibited a decreasing SOA concentration from marginal
seas of China to remote oceanic areas. In all, the model reproduced the SOA levels in the marine
atmosphere of the western Pacific Ocean reasonably well.

The comparison of the magnitudes between SOA and OA mass (1.4 times OC mass) concentrations

shown above indicates that SOA concentration was approximately 1~2 orders of magnitude lower than
OA over the western Pacific. Previous observation studies using the tracer-based approach also indicated
that the percentage contribution of SOA to OA was quite low over some marine areas (Fu et al., 2011;
Hu et al., 2013; Bikkina et al., 2014; Zhu et al., 2016). At Okinawa island, even considering all biogenic
sources (including isoprene, monoterpene, and sesquiterpene of both terrestrial and oceanic origins),
the measured concentration of total biogenic-SOA tracers was still less than 100 ng m$^{-3}$, with majority
of SOA tracers from local terrestrial biogenic emissions (Zhu et al., 2016). The above studies suggested
that primary organic aerosols was more important in remote marine atmosphere.

3.3 Aerosol optical depth





The model performance for aerosol optical depth (AOD) in the marine atmosphere of the western
Pacific was evaluated in this section. In-situ observations of AOD within the study domain were
obtained from the Aerosol Robotic Network (AERONET, https://aeronet.gsfc.nasa.gov/, last access:
2020/06/03). Level 2 AOD observations for the year 2014 were collected at the 6 coastal sites shown in
Figure 1. Hourly and monthly mean observations were derived from raw data and used for statistics
calculation and comparison. AOD at 550 nm was used to match the model output.
Figure 6 shows the temporal variations of the observed and simulated monthly mean AOD at the 6
AERONET sites. In general, RIEMS-Chem simulated the monthly mean AOD reasonably well in terms
of magnitude and monthly variation at almost all sites, although some biases occurred during some
months, such as the overpredictions in August at Fukuoka and in April at EPA-NCU, and the
underprediction in July at Yonsei University. For the sites in the northern oceanic areas (Ussuriysk,
Yonsei_University, Gwangju_GIST, and Fukuoka, Figure 6a~6d), both observations and simulations
generally exhibited higher AOD values in summer (JJA), moderately high AOD values from late winter
(JF) to spring (MAM), and relatively lower AOD values in autumn (SON). The simulated higher
inorganic aerosol concentrations in summer and late spring months could be responsible for the higher
AOD values in these regions. Besides, the higher relative humidity in summer due to the predominant
influence of maritime air masses also contributed to the maximum AOD values during summer months
(JJA) at these sites. On the other hand, for the sites in the southern oceanic areas (EPA-NCU and Chen-
Kung_Univ, Figure 6e and 6f), the monthly mean AOD was apparently higher from March to April and
remained low levels during the rest months. The above AOD peaks in spring could be attributed to the
continental outflows of biomass burning plumes originated from Southeast Asia, which were most
active in springtime in those regions (Hsiao et al., 2017; Tao et al., 2020). Table 6 shows the performance
statistics for hourly AOD at these AERONET sites. The overall annual mean AOD for the 6 sites was
0.34 from model simulation, which was very close to the observation of 0.37, with the NMB of -8%
and the overall correlation coefficient of 0.56 (0.41~0.67). The statistics indicate that the model was
able to reproduce aerosol optical properties over the western Pacific Ocean, which provides confidence
on the reliability of the subsequent estimation of aerosol radiative effect.

4 Model results
4.1 Marine primary organic and isoprene emissions





Figure 7a~7e show the estimated annual and seasonal mean MPOA emission rates over the western
Pacific of East Asia. In general, the spatial distribution of annual mean MPOA emission (Figure 7a)
resembled that of Chl-a (Figure 7f). The emission mainly occurred over two hotspot regions: the
marginal seas of China including the East China Sea, the Yellow Sea, and the Bohai Sea (EYB, denoted
in Figure 7a) and the northern parts of the western Pacific northeast of Japan (NWP, denoted in Figure
7a), with annual mean emission rates varying from $0.9\times10^{-2}$ μg m$^{-2}$ s$^{-1}$ to $1.8\times10^{-2}$ μg m$^{-2}$ s$^{-1}$. In SON,
high MPOA emission occurred in both the EYB and NWP regions, with the maximum up to $3.5 \times 10^{-2}$
μg m$^{-2}$ s$^{-1}$ in the NWP (Figure 7e), whereas MPOA emission was very low over the EYB in JJA (Figure
7d). The maximum seasonal mean emission rate of MPOA approached $3.6\times10^{-2}$ μg m$^{-2}$ s$^{-1}$ over the
Yellow Sea in DJF (Figure 7b), which was approximately 1/10 of the annual mean anthropogenic POA
emission rate in north China (on the order of $1.0\sim3.0\times10^{-1}$ μg m$^{-2}$ s$^{-1}$). Table 7 presents the seasonal and
annual averages of MPOA emission averaged over the western Pacific and the EYB and NWP regions.
In terms of oceanic average of the western Pacific, the mean MPOA emission generally exhibited the
largest emission rate in SON ($0.20\times10^{-2}$ μg m$^{-3}$ s$^{-1}$), moderately high emission rates in DJF ($0.18\times10^{-2}$
μg m$^{-2}$ s$^{-1}$) and MAM ($0.17\times10^{-2}$ μg m$^{-2}$ s$^{-1}$), and the lowest one in JJA ($0.08\times10^{-2}$ μg m$^{-2}$ s$^{-1}$), with an
annual average of $0.16\times10^{-2}$ μg m$^{-2}$ s$^{-1}$ (Table 7). It is interesting to note that the seasonal variation of
MPOA emission was not consistent with that of Chl-a concentration, which exhibited higher values in
SON and JJA and the lowest one in DJF (Table 7). This is because MPOA emission rate is determined
by the combined effect of Chl-a concentration and sea salt emission flux, and sea salt flux is mainly
controlled by surface wind speed according to the scheme in section 2.3.1. In terms of seasonal and
domain average over the western Pacific, the maximum Chl-a concentration and the second largest sea
salt emission flux in SON led to the largest MPOA emission in autumn (Table 7). However, although
Chl-a concentration was also high in JJA (1.07 mg m$^{-3}$, Table 7), the sea salt flux was the minimum in
JJA (0.14 μg m$^{-2}$ s$^{-1}$, Table 7) due to the weakest wind speed (3.0 m s$^{-1}$, Table 9), resulting in the lowest
MPOA emission in summer (Table 7). Although the sea salt emission flux reached the maximum in DJF
(Table 7) due to the largest wind speed in this season (Table 9), the winter Chl-a concentration was
lowest, leading to a moderate MPOA emission in winter (Table 7), in a similar magnitude to that in
spring when moderately high Chl-a concentration and relatively low sea salt flux occurred. In all, the
MPOA emission rate over the western Pacific exhibited an apparent seasonality of SON > DJF ≈ MAM >
JJA.





For the EYB region, the maximum MPOA emission occurred in winter (DJF) (Figure 7b and Table
7) with a seasonal and domain average of $1.2\times10^{-2}$ µg m$^{-2}$ s$^{-1}$, which was 10 times larger than the
minimum of $0.12\times10^{-2}$ µg m$^{-2}$ s$^{-1}$ in summer (JJA) (Figure 7d and Table 7). Although Chl-a
concentrations were similar between DJF and JJA, the sea salt flux in DJF was approximately 9 times
that in JJA (Table 7). So, the seasonality of MPOA emission in the EYB region was mainly determined
by that of sea salt emission flux due to the weak seasonal variation of Chl-a concentration. Differently,
in the NWP region, MPOA emission exhibited the maximum value in SON, followed by those in MAM
and DJF, and the lowest ones in JJA (Table 7). It is interesting to note that although both the Chl-a
concentration and sea salt emission flux were slightly higher in MAM than those in SON, the MPOA
emission was higher in SON. This could be explained that high Chl-a concentration and large sea salt
emission flux often occurred simultaneously in SON, strengthening the MPOA emission flux, whereas
in MAM, the times of high Chl-a concentration and large sea salt flux often mismatched, reducing the
MPOA emission. The MPOA emissions in winter and summer were in a similar level in the NWP region,
about 40% lower than that in autumn.
The distribution pattern of MPOA emission in the western Pacific from this study is similar to those
from previous model studies (Spracklen et al., 2008; Gantt et al., 2009; Huang et al., 2018), but the
magnitude of the simulated MPOA emission flux is larger than previous estimates. For example, the
annual mean MPOA emission rates over the western Pacific were estimated to vary from 0.1 to
approximately 12 ng m$^{-2}$ s$^{-1}$ in previous studies (Spracklen et al., 2008; Vignati et al., 2010; Gantt et al.,
2011; Long et al., 2011; Huang et al., 2018), whereas the estimates in this study ranged from 3 to 18 ng
m$^{-2}$ s$^{-1}$ (Figure 7a). The larger marine POA emission estimated in this study could be attributed to the
application of the daily mean Chl-a concentration from satellite retrievals and of a finer model grid
resolution (60 km) compared with those in global models. On average, the annual MPOA emission was
estimated to be 0.78 Tg yr$^{-1}$ over the western Pacific (with an ocean area of $1.58\times10^{7}$ km$^{2}$) from this
study. For comparison, the global annual emission of sub-micron MPOA was estimated to vary from
6.3 Tg yr$^{-1}$ to 9.4 Tg yr$^{-1}$ based on different parameterizations, models, and study periods (Vignati et al.,
2010; Meskhidze et al., 2011; Gantt et al., 2012a; Huang et al., 2018). This suggests the western Pacific
of East Asia contributed 8~12% of the global annual MPOA emission with only 4.4% of the global
ocean area (approximately $3.6\times10^{8}$ km$^{2}$). The regions of EYB and NWP comprised approximately 2%
and 18% of the western Pacific in terms of area, respectively, but they contributed 8% and 46% of the



MPOA emission in terms of annual mean. This study revealed that the EYB and NWP are important
bloom regions, accounting for more than half of the total MPOA emission over the western Pacific.
Table S2 presents the simulated marine isoprene emission fluxes in comparison with observation-
based estimates over the western Pacific of East Asia and other oceans from previous studies. Over the
western North Pacific, the observed marine isoprene emission flux showed larger values in May
(140~143.8 nmol m$^{-2}$ day$^{-1}$), a moderate value in August (~55.6 nmol m$^{-2}$ day$^{-1}$), and the lowest one in
winter (~21.4 nmol m$^{-2}$ day$^{-1}$). The model simulation generally agreed with observation in terms of both
seasonality and magnitude except for the low bias in spring (85~89 nmol m$^{-2}$ day$^{-1}$ in spring, ~63 nmol
m$^{-2}$ day$^{-1}$ in summer, and ~26 nmol m$^{-2}$ day$^{-1}$ in winter), which could be associated with the different
years. According to equations (2) and (3), both Chl-a concentration and incoming solar radiation
determine marine biogenic VOCs emission, the larger isoprene flux in May was mainly due to the
maximum Chl-a concentration in spring over the NWP region (Table 7). It is interesting to note that the
marine isoprene emission flux in May from Matsunaga et al. (2002) was in a similar magnitude to that
from Ooki et al. (2005) over the western Pacific despite in different years (Table S1). Over the marginal
seas of China, Li J. L. et al. (2017; 2018) observed higher marine isoprene emission flux in July-August
(~161.5 nmol m$^{-2}$ day$^{-1}$) than those in October-November (~48.3 nmol m$^{-2}$ day$^{-1}$) and May-June (~36.1
nmol m$^{-2}$ day$^{-1}$) during 2013-2014. The model well reproduced the seasonal trend and magnitude of
isoprene flux, with corresponding mean values of 130 nmol m$^{-2}$ day$^{-1}$, 48 nmol m$^{-2}$ day$^{-1}$, and 35 nmol
m$^{-2}$ day$^{-1}$ during the same periods of 2014, respectively. The apparently higher isoprene flux in July-
August was mainly resulted from the strongest solar radiation in summer, although the Chl-a
concentration was not highest in this season in the EYB region (Table 7). Table S2 also lists previously
observed marine isoprene emission fluxes over the Southern Ocean and Arctic Ocean in summer for
reference. The domain-wide annual marine isoprene emission estimated over the western Pacific was
0.015 Tg yr$^{-1}$ in this study. Arnold et al. (2009) calculated with GEOS-Chem model a global-annual
isoprene emission of 0.31 Tg yr$^{-1}$. This suggests that the western Pacific region contributed
approximately 5% of the global marine isoprene emission, although different model and study period
are selected. However, some previous studies (Arnold et al., 2009; Booge et al., 2016) found the
emission flux calculated by current marine isoprene emission schemes tended to yield lower isoprene
concentration in marine atmospheres compared with observations.



### 4.2 Marine organic aerosols and their relative importance

Annual and seasonal mean near surface MOA concentrations, MSOA concentrations, and the percentage contributions of MOA to total OA mass in the study domain were shown in Figure 8. The spatial distributions of MOA concentrations (Figure 8a~8e) generally resembled those of MPOA emissions (Figure 7a~7e). It is remarkable that MPOA concentration (MOA minus MSOA) was approximately 1~2 orders of magnitude higher than MSOA concentration (with concentration of several ng m$^{-3}$) in the western Pacific (Figure 8a~8e vs Figure 8f~8j), indicating that MPOA constituted a dominant fraction of MOA, which will be discussed below. Figure 8a shows that high MOA concentrations mainly occurred over the EYB and NWP regions, with the annual and regional averages being 0.48 μg m$^{-3}$ and 0.59 μg m$^{-3}$, respectively (Table 8), accounting for 13% (6~30%) and 42% (30~60%) of total OA mass in these two regions, respectively (Figure 8k and Table 8). The larger MOA contribution over the NWP was attributed to the high MOA level and the relatively low total OA level there. It is noticed that MOA even influenced the coastal areas of eastern China. The annual mean MOA concentration decreased from approximately 0.5 μg m$^{-3}$ in coastal areas to 0.1 μg m$^{-3}$ in the inland areas (Figure 8a), accounting for approximately 2% to 6% of the near-surface OA mass in the coastal regions (Figure 8k). The maximum seasonal mean MOA concentration over the coastal areas of eastern China could be up to 0.6 μg m$^{-3}$ to 0.8 μg m$^{-3}$ in MAM (Figure 8c) and SON (Figure 8e). The domain and seasonal mean MOA concentration over the western Pacific exhibited the maximum value in MAM (0.37 μg m$^{-3}$), follow by that in SON (0.26 μg m$^{-3}$), and relatively lower concentrations in JJA (0.23 μg m$^{-3}$) and DJF (0.21 μg m$^{-3}$) (Table 8). It was noteworthy that the seasonality of MOA concentration was different from that of MPOA emission, which could be attributed to the influence of different meteorological conditions and physical processes. In the western Pacific, although MPOA emission peaked in SON (Table 7), MOA concentration peaked in MAM (Table 8). It is noticed that precipitation was lowest and wind speed was low in MAM (Figure 9c and 9h, Table 9), leading to a smaller dry deposition velocity (Zhang et al. 2001) and the weakest wet scavenging, both favored accumulation of MOA and thus resulted in the highest MOA level in spring. On the contrary, due to the maximum wind speed and relatively more precipitation in DJF (Figure 9b and 9g, Table 9), the mean MOA concentration was lowest in winter.

For the EYB region, northwesterly winds prevailed In DJF and SON and turned to northeasterly winds over marginal seas of southeast China (Figure 9b and 9e), which transported MOA from the major





MPOA source region (EYB) to the northern part of the South China Sea (Figure 8b and 8e). As wind
speed over the EYB was low in MAM and JJA (Figure 9c and 9d, Table 9), MOA was mainly restricted
within this region (Figure 8c and 8d). In terms of seasonal average, MOA concentration experienced its
maximum in MAM, followed by those in DJF and SON, and the minimum in JJA (Figure 8b~8e). The
seasonal and regional mean MOA concentrations over the EYB were 0.62 μg m$^{-3}$, 0.54 μg m$^{-3}$, 0.52 μg
m$^{-3}$, and 0.22 μg m$^{-3}$ for MAM, DJF, SON, and JJA, respectively (Table 8). The different seasonality
between MOA concentration (Table 8) and MPOA emission (Table 7) in the EYB region could also be
mainly attributed to meteorological conditions. The MPOA emission was relatively low in MAM (Table
7), but the second lowest wind speed and less precipitation (Table 9) favored aerosol accumulation,
resulting in the highest MOA concentration in spring (Table 8). The minimum MPOA emission and the
maximum precipitation in JJA led to the minimum MOA concentration in summer. Although MPOA
emission was largest in SON and DJF (Table 7), the maximum wind speeds (Table 9) led to stronger
dry deposition of aerosols and thus a moderate MOA concentration in the two seasons (Table 8).
MOA concentration over the NWP region exhibited apparent higher concentrations in MAM and
JJA than those in SON and DJF (Figure 8b~8e), with the regional and seasonal averages reaching 0.81
μg m$^{-3}$ in MAM, 0.80 μg m$^{-3}$ in JJA, 0.52 μg m$^{-3}$ in SON, and 0.23 μg m$^{-3}$ in DJF, respectively (Table
8). Using GEOS-Chem with a different marine organic aerosol emission scheme, Spracklen et al. (2008)
also showed that in the North Atlantic along the similar latitude bands to the NWP (~35°N to ~55°N),
both observation and simulation exhibited higher OC concentrations in summer and spring than in the
other seasons at the Azores Island and Mace Head Island. The strong seasonality of MOA over the NWP
was also attributed to the combined effects of MPOA emission, wind speed, and precipitation. In MAM,
the high MOA concentration over the NWP was mainly due to the large MPOA emission (Figure 7c and
Table 7), which was just smaller than that in SON, and partly due to the relatively weak dry deposition
and wet scavenging caused by moderate wind speed and precipitation in this season (Table 9). In JJA,
although the MPOA emission was small, the lowest wind speed and precipitation in JJA over the NWP
(2.5 m s$^{-1}$ and 3.7 cm grid$^{-1}$ month$^{-1}$, Table 9) led to the weakest dry deposition and wet scavenging of
particles in summer, resulting in a long residence time of MOA and consequently the high MOA
concentration in summer over the NWP. In SON, although the MPOA emission was largest over the
NWP (Table 7), the mean wind speed was high over the northern part of the NWP (Figure 9e) where
MPOA emission mainly occurred (Figure 7e), leading to strong dilution of MOA particles in autumn.



Furthermore, the secondly largest precipitation over the NWP in SON (Table 9) caused strong wet
scavenging of particles, also contributed to the relatively low MOA level. In DJF, the wind speed was
largest, about 2 times those in the other seasons, and the precipitation was also the maximum (Table 9,
Figure 9b and 9g), leading to the lowest MOA concentration in winter over the NWP (Figure 8b and
Table 8).

As shown in Figures 8k~8o, MOA generally accounted for approximately 30% to over 60% of total

OA concentration over the remote oceans of high (>35°N) and low (<25°N) latitudes. The large
MOA/OA ratios over the remote oceans of high latitude (including NWP) could be attributed to the high
MOA concentration due to large marine emissions there; whereas, the large MOA/OA ratios over the
subtropical oceans of low latitude were mainly due to the low total OA level (small denominator).
Averaged over the NWP region, the annual mean MOA/OA ratio was 42%, with higher contributions
in MAM (52%) and SON (48%) and lower ones in DJF (36%) and JJA (32%) (Table 8). Although MOA
concentration over the NWP was secondly highest in JJA, its contribution was small because OA
transported from land sources also subject to weak dry deposition and wet scavenging, which led to
higher OA level and lower MOA/OA ratio. Over the EYB region, MOA accounted for approximately
6% to 30% of the total OA in terms of annual mean (Figures 8k). In terms of annual and regional average,
the MOA/OA ratio was 13%, with higher ratios in SON (18%) and MAM (15%), a moderate one in
DJF (11%), and the lowest one in JJA (6%) (Table 8), similar to the seasonality over the NWP. It was
impressive that the importance of MOA in total OA increased as the distance to the East Asian continent
increased over the western Pacific. It is also interesting to note that MOA even accounted for
approximately 2~6% of the annual mean OA mass over portions of southeast China (Figures 8k), and
such contribution could be as high as 8~10% in the coastal areas in SON (Figures 8o) and MAM
(Figures 8m).

It all, both the MOA concentration and the MOA contribution to total OA were lowest in summer

(JJA) in the EYB region, which was mainly due to the much smaller MPOA emission in this season.
However, in the NWP region, although the MPOA emission was also lowest in summer, MOA
concentration in summer was in a same level as that in spring, and larger than those in the other seasons,
because dry deposition velocity and precipitation were lowest in summer, which favored aerosol
accumulation and a high level of MOA.

SOA produced by marine biogenic VOCs (isoprene and terpene) was on the order of $10^{-2}$~$10^{-3}$ μg



m$^{-3}$ (Figure 8f~8j), which was much lower than the MPOA concentration. The spatial distribution of
MSOA exhibited high concentrations over the EYB and NWP regions in terms of annual mean, with
values up to 6 ng m$^{-3}$ (approximately 0.5% of MOA concentration) over these two regions (Figure 8f).
MSOA concentration exhibited the maximum in JJA, with seasonal mean values of ~7 ng m$^{-3}$ to ~11 ng
m$^{-3}$ extending from the marginal seas of China (EYB) to remote western North Pacific (NWP) (Figure
8i). MSOA distribution in MAM was similar to that in JJA but with lower mean concentrations (4~7 ng
m$^{-3}$) over the EYB and NWP regions (Figure 8h). In SON (Figure 8j), MSOA concentrations were 2~4
ng m$^{-3}$ in the above two regions. In DJF (Figure 8g), MSOA concentration was lowest, with values of
0.4~2 ng m$^{-3}$ over the marginal seas of China and the southern parts of the western Pacific. The
maximum seasonal mean MSOA concentration was up to 14 ng m$^{-3}$ over oceanic areas of the EYB to
NWP regions in JJA, and the maximum daily mean MSOA value exceeded 28 ng m$^{-3}$ on some days,
e.g. June 6~7 (figure not shown). Table 8 shows the domain and seasonal/annual averages of MSOA
over the oceanic regions of concern. The annual mean MSOA concentrations were 2.2 ng m$^{-3}$, 4.1 ng
m$^{-3}$ and 3.8 ng m$^{-3}$ averaged over the western Pacific, the EYB and NWP regions. It is striking that the
domain average MSOA concentration consistently exhibited a distinct seasonality, with the maximum
in summer and the minimum in winter throughout all the oceanic regions of the western Pacific, which
was resulted from the combined effects of isoprene emission flux and meteorological conditions. The
domain average MSOA concentrations reached the maximums of 3.9 ng m$^{-3}$, 7.5 ng m$^{-3}$, and 8.3 ng m$^{-}$
$^{3}$, respectively, over the western Pacific, the EYB and NWP regions in JJA. The seasonality of MSOA
concentration over the western Pacific is similar to the simulation result from Myriokefalitakis et al.
(2010). According to Table 8, the annual mean fraction of MSOA in MOA was estimated to be 0.8%,
0.9% and 0.6%, over the western Pacific, the EYB and NWP regions, respectively. The maximum and
minimum fractions of MSOA in MOA averaged over the western Pacific occurred in JJA (1.7%) and
DJF (0.3%), respectively, with the maximum regional and seasonal average MSOA fraction up to 3.4%
in summer over the EYB region. Based on the GEOS-Chem model simulation, Arnold et al. (2009)
indicated that SOA produced by marine isoprene contributed only a very small fraction (0.01~1.4%) of
the observed organic aerosol mass at remote marine sites (Amsterdam Island in southern Indian Ocean,
Azores and Mace Head islands in northern Atlantic Ocean). In a global model simulation from
Myriokefalitakis et al. (2010), the annual mean marine isoprene and monoterpene derived SOA
concentrations were approximately 0.4~1 ng m$^{-3}$ (accounting for ~0.4% of marine OA) over the western



Pacific. Meskhidze et al. (2011) illustrated the marine SOA from phytoplankton-derived isoprene and
monoterpenes contributed <10% of surface OM concentration of marine source in most areas of the
western Pacific.

4.3 Direct radiative effect due to MOA

In this section, the direct radiative effect (DRE) due to MOA ($DRE_{MOA}$) over the western Pacific of

East Asia was analyzed and estimated. The $DRE_{MOA}$ was derived by subtracting the model result of the
NoMOE case from that of the FULL case.

Figures 10a to 10e show the annual and seasonal mean $DRE_{MOA}$ at TOA under all-sky condition.

MOA induced an annual mean DRE of -0.1 ~ -0.9 W m$^{-2}$ over the western Pacific (Figure 10a).
Consistent with the spatial distribution of MOA concentration, the maximum $DRE_{MOA}$ (-0.9 W m$^{-2}$)
occurred over the NWP region (Figure 10a). Over the EYB region, the other hotspot of MOA mass
concentration, the $DRE_{MOA}$ was weaker, with an annual mean $DRE_{MOA}$ of -0.2 ~ -0.5 W m$^{-2}$ (Figure
10a). In terms of domain average, the annual mean $DRE_{MOA}$ was estimated to be -0.21 W m$^{-2}$ over the
western Pacific, smaller than that over the NWP (-0.41 W m$^{-2}$) but similar to that over the EYB (-0.24
W m$^{-2}$) (Table 10). The annual mean $DRE_{MOA}$ over the western Pacific from this study was stronger
than the global mean $DRE_{MOA}$ at TOA (-0.16 W m$^{-2}$) estimated based on a 10-yr model simulation from
Huang et al. (2018). The mean $DRE_{MOA}$ over the western Pacific was largest in spring (-0.31 W m$^{-2}$)
and lowest in winter (-0.14 W m$^{-2}$) (Table 10), consistent with the seasonality of MOA concentration.

For the NWP region, MOA induced the largest all-sky DRE in MAM (-0.6 ~ -1.6 W m$^{-2}$) (Figure

10c) and followed by that in JJA (-0.5 ~ -1.3 W m$^{-2}$) (Figure 10d) mainly due to higher MOA
concentrations in the two seasons. The $DRE_{MOA}$ value was relatively low in SON (-0.3 ~ -0.6 W m$^{-2}$)
(Figure 10e), and it was lowest in DJF, with the maximum of just -0.4 m$^{-2}$ (Figure 10b) due to the lowest
MOA concentration in winter (Table 8). The regional and seasonal averages of $DRE_{MOA}$ over the NWP
were estimated to be -0.68 W m$^{-2}$, -0.58 W m$^{-2}$, -0.23 W m$^{-2}$, and -0.16 W m$^{-2}$ in MAM, JJA, SON, and
DJF, respectively (Table 10). On the contrary, the $DRE_{MOA}$ over the EYB region exhibited a different
seasonal trend from that over the NWP, exhibiting the largest DRE in SON (Figure 10e), moderate
DREs in DJF (Figure 10b) and MAM (Figure 10c), and the lowest one in JJA (Figure 10d), with
corresponding mean values of -0.28 W m$^{-2}$, -0.25 W m$^{-2}$, -0.24 W m$^{-2}$, and -0.17 W m$^{-2}$, respectively,
for the four seasons (Table 10). The weaker $DRE_{MOA}$ over the EYB (-0.24 W m$^{-2}$ in terms of annual



mean) than that over the NWP (-0.41 W m$^{-2}$) could be attributed to both the lower MOA concentration
(Table 8) and lower relative humidity (73% vs 83%, Table 9).
It is of interest to estimate the relative importance of MOA in directly perturbing solar radiation
compared with that of total aerosols over the western Pacific. Table 10 lists the simulated annual and
seasonal mean DREs due to all aerosols (the sum of all anthropogenic aerosols, mineral dust, and marine
aerosols) over the western Pacific and the regions of NWP and EYB, respectively. It is remarkable that
the DRE$_{MOA}$ was quite small compared with that due to all aerosols throughout the western Pacific.
Over the EYB region, the DRE$_{MOA}$ was almost negligible because of the predominant influence of
anthropogenic emissions. Over the remote oceans (NWP), although the absolute value of DRE$_{MOA}$ was
still small, its relative importance increased due to weakened DREs by anthropogenic aerosols. Under
all-sky condition, the annual mean DRE$_{MOA}$ averaged over the western Pacific and the NWP were
approximately 1.2% and 2.3% of the DREs due to all aerosols, respectively, with the seasonal means
varying from 0.9% to 1.4% over the western Pacific, 0.5% to 1.6% over the EYB, and 1.6% to 3.5%
over the NWP. In all, MOA plays a minor role in directly affecting solar radiation over the western
Pacific of East Asia.
It should be mentioned that due to the much smaller MSOA concentration than MPOA
concentration, the above DRE$_{MOA}$ was dominantly contributed by MPOA, similar to the findings from
previous studies (Arnold et al., 2009; Booge et al., 2016; Li et al., 2019).

4.4 Indirect radiative effect due to marine organic aerosols
The indirect radiative effect (IRE) due to marine organic aerosols (IRE$_{MOA}$) over the western Pacific
of East Asia was explored in this section. The annual and seasonal mean IRE$_{MOA}$ at TOA are shown in
Figure 10f to 10j. The IRE$_{MOA}$ was negative, resulting from a series of changes in cloud properties
induced by MOA, i.e. an increase in cloud droplet number concentration, a decrease in cloud droplet
effective radius, an increase in cloud optical depth and cloud water path, a decrease in cloud water to
rain water conversion, and consequently more reflection of solar radiation out of the TOA. The model
simulated cloud properties have been compared against satellite retrievals in spring 2014 in our previous
study (Han et al., 2019), which indicated the model was able to reasonably reproduce the major features
in cloud property distribution. It is remarkable that the IRE$_{MOA}$ was apparently stronger than the
DRE$_{MOA}$ over the western Pacific, with the maximum annual mean of IRE$_{MOA}$ more than 10 times that





of DRE$_{MOA}$, although the positions of their maximum values were different. The annual mean IRE$_{MOA}$
of -4 ~ -12 W m$^{-2}$ distributed from southwest to northeast over wide areas of the western Pacific (Figure
10f). It is evident that the strongest IRE$_{MOA}$ occurred in DJF, with the seasonal mean values of -8 ~ -14
W m$^{-2}$ over vast areas from the East China Sea to the oceans east of Japan (Figure 10g). The IRE$_{MOA}$ in
MAM was similar in distribution pattern to that in DJF, with lower values of -8 ~ -10 W m$^{-2}$ from the
East China Sea to the oceans south of Japan (Figure 10h). The IRE$_{MOA}$ was weakest in JJA, with the
maximum of -6 W m$^{-2}$ over a portion of the western Pacific east of Japan (Figure 10i), whereas the
IRE$_{MOA}$ value in SON was between those in MAM and JJA, with a similar distribution pattern. The
seasonal variation of IRE$_{MOA}$ was likely influenced by both the seasonal changes in cloud amount and
MOA concentration. In terms of domain average, the seasonal mean IRE$_{MOA}$ was strongest (-6.0 W m$^{-}$
$^{2}$) in DJF over the western Pacific (Table 10), which could be mainly due to the largest cloud fraction
in DJF (Figure S1g), although MOA concentration was lower in winter (Table 8). On the contrary,
although MOA concentration reached the maximum in MAM, because cloud fraction was relatively
lower (Figure S1h) in this season, the IRE$_{MOA}$ was secondly largest in spring. The weakest IRE$_{MOA}$
occurred in JJA, which was mainly attributed to both the lower MOA concentration and cloud fraction
in summer (Table 8, Figure S1i). In springtime when MOA concentration was highest over the western
Pacific (Table 8), the domain and seasonal mean IRE$_{MOA}$ can be as high as -14.8 W m$^{-2}$ (Table 10).
Figure S2 further presents the monthly mean Chl-a concentration, MPOA emission, MOA concentration,
and IRE$_{MOA}$ in April, when Chl-a concentration and MPOA emission resulting from phytoplankton were
distinctly high in the EYB and NWP regions (Figure S2a and S2b). It can be found that MOA was
transported from the high Chl-a regions to the south under north or northwesterly winds over the oceans
(Figure S2c), resulting in an elevated IRE$_{MOA}$ up to -18 W m$^{-2}$ over the western Pacific south and east
of Japan. Previous studies are very limited to compare with. Our simulated IRE$_{MOA}$ in the NWP was in
a similar magnitude to that in Meskhidze and Nenes (2006), which estimated based on satellite retrievals
a reduction of 15 W m$^{-2}$ in shortwave radiation at TOA due to changes in cloud properties during a
strong phytoplankton bloom near South Georgia Island in the Southern Ocean in summertime. Gantt et
al. (2012b) estimated a 10-year average shortwave cloud forcing of approximately ~ -5 Wm$^{-2}$ due to
marine organic aerosols in the western Pacific Ocean by using a global model CAM5 with 1.9°×2.5°
horizontal grid resolution. The maximum annual mean IRE estimated in this study can be -12.1 Wm$^{-2}$
(Table 10) over the western Pacific, apparently stronger than that from Gantt et al. (2012b), which could





be due to the use of a regional model with finer grid resolution, daily Chl-a satellite data, and the
different study period.
The annual and regional mean $IRE_{MOA}$ was estimated to be -4.2 W m$^{-2}$ for the western Pacific, -2.2
W m$^{-2}$ over the EYB region, and -4.1 W m$^{-2}$ over the NWP region, respectively (Table 10). There was
an apparent seasonality in the $IRE_{MOA}$, with the maximum of -6.0 W m$^{-2}$ in DJF and the minimum of -
1.9 W m$^{-2}$ in JJA over the western Pacific (Table 10). However, the seasonality of $IRE_{MOA}$ in the EYB
and NWP regions are different from that over the western Pacific. Over the EYB, the estimated $IRE_{MOA}$
reached the maximum (-2.9 W m$^{-2}$) in SON, which was due to the combined effect of a moderately high
MOA concentration (Table 8) and the maximum cloud fraction (Figure S1j) in this region. The $IRE_{MOA}$
was in a range of -1.5 ~ -2.4 W m$^{-2}$ in other seasons. Although MOA concentration reached the
maximum in MAM, there was a minimum total cloud fraction in spring among seasons (Figure S1h),
leading to a moderate $IRE_{MOA}$. For the NWP region, the $IRE_{MOA}$ in JJA (-2.5 W m$^{-2}$) was remarkably
smaller than those in other seasons (-4.0 ~ -5.1 W m$^{-2}$) and $IRE_{MOA}$ reached the maximum in MAM (-
5.1 W m$^{-2}$), which was mainly due to the maximum MOA concentration in spring (Table 8). The weakest
$IRE_{MOA}$ in JJA was mainly attributed to the lower cloud fraction in summer (Table 8, Figure S1i).
The relative importance of MOA in the aerosol indirect radiative effect over the western Pacific
was investigated by comparing the $IRE_{MOA}$ with the IREs induced by sea salt and all aerosols. In terms
of annual and oceanic average, the IREs due to sea salt and all aerosols were estimated to be -3.7 W m$^{-}$
$^{2}$ and -13.3 W m$^{-2}$ (Table 10), respectively, which means the $IRE_{MOA}$ (-4.2 W m$^{-2}$) was comparable to
the IRE by sea salt and approximately 32% of that by all aerosols. It is noteworthy that the relative
importance of MOA was strengthened over the regions of EYB and NWP, accounting for approximately
42% and 36% of the IRE due to all aerosols. In terms of seasonal and domain average over the western
Pacific, the $IRE_{MOA}$ was approximately 31-38% of the IRE by all aerosols in seasons except JJA, and
20% in JJA. The above model estimation demonstrates that the indirect radiative effect due to MOA can
be approximately one third (32%) of the IRE due to all aerosols, suggesting an important role of MOA
in perturbing radiation transfer through modifying cloud properties over the western Pacific Ocean of
East Asia.
It is interesting to found that the estimated IRE by MSOA (note assuming external mixing with sea
salt) accounted for approximately 6.4% of the annual mean $IRE_{MOA}$ averaged over the western Pacific
(table not shown), although the annual mean MSOA concentration was approximately 0.8% of the MOA



concentration (Table 8), and the percentage contribution of MSOA to the $IRE_{MOA}$ increased to 13.7% in
JJA, consistent with the maximum fraction of MSOA in MOA in summer (1.7%) over the western
Pacific (Table 8). As for the EYB and NWP regions, the maximum contribution of MSOA to the $IRE_{MOA}$
both occurred in summer, with the percentage contributions of 11.8% and 17.7%, respectively, whereas
the MSOA contribution was less than 10% in other seasons, and was smallest in winter (1~3% in the
two regions), consistent with the seasonal variation of MSOA concentration. Overall, MSOA plays a
minor role in perturbing cloud properties and shortwave radiation compared with MPOA.

5. Conclusions.
The organic aerosols of marine origin over the western Pacific Ocean of East Asia was investigated
by an online-coupled regional climate-chemistry model RIEMS-Chem for the year 2014. Emissions and
relevant processes of marine MPOA, isoprene and monoterpene were incorporated into RIEMS-Chem.
A wide variety of observational datasets from EANET, CNEMC and AERONET networks, cruise
measurements and previous publications were collected for model validation. The modeled SOA from
marine VOC sources was also compared with secondary organic tracers measured by research cruise.
The model performed well for $PM_{2.5}$ and $PM_{10}$ in marine environment, producing overall correlation
coefficients and NMBs of 0.61/0.70 and 12%/-7% for $PM_{2.5}$ concentration, 0.65/0.65 and -5%/-1% for
$PM_{10}$ concentration at the EANET/CNEMC sites, respectively. The model reasonably reproduced the
spatial distribution and temporal variation of BC and OC concentrations along cruise tracks and at
islands over the west Pacific, with correlation coefficients and NMBs being in the range of 0.79~0.88
and 10%~18% for BC and 0.6~0.75 and -28%~3% for OC, respectively. The modeled OC concentration
was apparently improved while taking into account marine organic aerosols. The model result clearly
showed an increasing contribution of marine organic aerosols to total OC mass concentration from the
marginal seas of China to remote oceans. Organic aerosol mass of marine origin were dominated by
MPOA because MSOA produced by marine isoprene and monoterpene emissions was about 1~2 orders
of magnitude lower than MPOA. The model performance for AOD at the 6 coastal AERONET sites was
reasonably well, with an overall correlation coefficient of 0.56 and an NMB of -8%.
High MPOA emission mainly occurred over the marginal seas of China (EYB) and the northern
parts of western Pacific northeast of Japan (NWP). For the western Pacific, MPOA emission reached
the maximum in SON, followed by those in DJF and MAM, and the minimum in JJA, with an annual





and domain average emission rate of $0.16 \times 10^{-2}$ μg m$^{-2}$ s$^{-1}$. The combination of Chl-a concentration and
sea salt emission flux determined the seasonality of MPOA emission. The annual MPOA emission for
the year 2014 was estimated to be 0.78 Tg yr$^{-1}$ over the western Pacific, which might account for
approximately 8~12% of global annual MPOA emission.
Consistent with the distribution pattern MPOA emission, high MOA concentration mainly
distributed over the EYB and NWP, with an annual and domain mean concentration of 0.27 μg m$^{-3}$, 0.48
μg m$^{-3}$ and 0.59 μg m$^{-3}$, over the western Pacific, the EYB and NWP regions, respectively. MOA
concentration was highest in MAM and lowest in DJF, with the seasonal and domain mean values of
0.37 μg m$^{-3}$ and 0.21 μg m$^{-3}$, respectively, over the western Pacific. The seasonality of MOA
concentration was determined by the combined effect of MPOA emission, dry and wet depositions.
On average, the annual mean percentage contribution of MOA to total OA mass was 26% over the
western Pacific, with the largest seasonal mean contribution of 32% in SON and the lower ones in DJF
(24%) and JJA (23%). Over the NWP, the domain average contribution of MOA to OA could be as high
as 42% in terms of annual mean and approaching 52% in MAM; however, over the EYB, the annual
mean contribution was just 13% and the percentage contribution was even reduced to 6% in JJA. This
indicated that the relative importance of MOA in total OA concentration increased with the distance
away from the East Asian continent. MSOA concentration was approximately 1~2 orders of magnitude
lower than MPOA, with the simulated annual and regional mean MSOA being 2.2 ng m$^{-3}$ and the
maximum daily mean value up to 28 ng m$^{-3}$ in summer over the western Pacific.
MOA had a minor impact on aerosol direct radiative effect over the western Pacific, with an
annual/domain mean all-sky DRE$_{MOA}$ at TOA being -0.21 W m$^{-2}$ (approximately 1.2% of the DRE due
to all aerosols). On the contrary, MOA exerted a considerable indirect radiative effect. The annual and
domain mean IRE$_{MOA}$ was estimated to be -4.2 W m$^{-2}$ over the western Pacific, with the maximum in
winter (-6.0 W m$^{-2}$) and the minimum in summer (-1.9 W m$^{-2}$) and the monthly mean IRE$_{MOA}$ can reach
-18 W m$^{-2}$ in April. The changes in MOA concentration and cloud amount both contributed to the
seasonality of IRE$_{MOA}$. In terms of annual and regional mean over the western Pacific, MSOA just
contributed approximately 6% of the IRE$_{MOA}$, which meant MPOA dominated the IRE$_{MOA}$. The mean
IRE$_{MOA}$ was approximately 32% of the IRE due to all aerosols, which indicated MOA had a considerable
impact on aerosol indirect radiative effect over the western Pacific.
While this study presents new aspects on seasonal variation and annual means of emissions,



concentrations, and radiative effects of MOA in the western Pacific, it is still subject to some
uncertainties as follows: 1.) the properties of marine organic aerosols, including size distribution,
molecular weight, solubility, surfactant amount etc. are still poorly characterized, which are crucial to
aerosol activation, dry deposition, and wet scavenging; 2.) the sources and chemical formation processes
of marine secondary organic aerosols are highly complex, and poorly understood and represented in the
model; 3.) the indirect effects of MOA in this study is for warm stratiform cloud. Further research on
MOA sources, properties, and chemical processes will be conducted together with the advances in both
field experiments and model development in the future.


**Author Contributions.**
ZH designed the study, JL and ZH developed the model, processed and analyzed the model results, JL
performed the model simulation, ZH and JL wrote the paper, PF and XY provided and analyzed the
cruise measurement data.

**Data availability.**
The observational data can be accessed through contacting the corresponding author.

**Competing interests.**
The authors declare that they have no conflict of interests.

**Special issue statement.**
This article is part of the special issue "Marine organic matter: from biological production in the ocean
to organic aerosol particles and marine clouds". It is not associated with a conference.

**Acknowledgement.**
This study was supported by the National Key R&D Program of China (2019YFA0606802) and the
Jiangsu Collaborative Innovation Center for Climate Change.

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

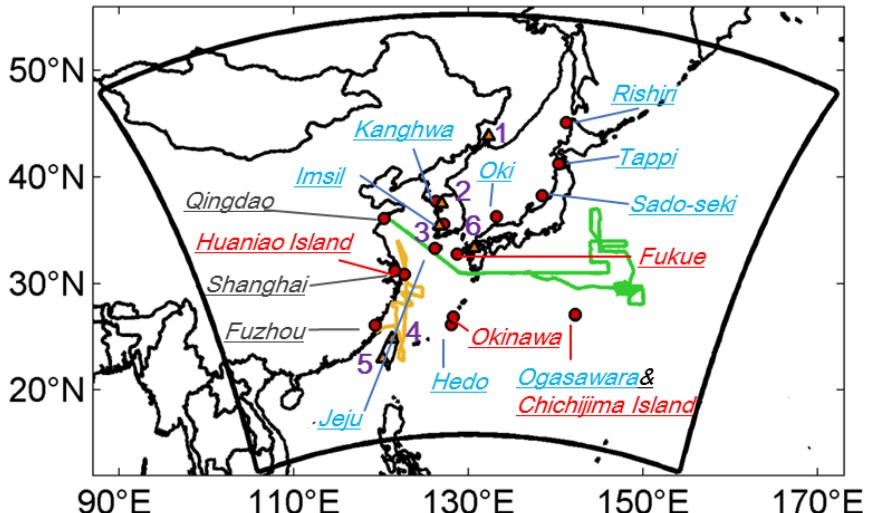

Figure 1. Model domain, observational sites, and research cruise tracks. EANET sites are marked in light-blue. Observation sites of carbonaceous aerosols are marked in red (Chichijima Island: Boreddy et al., 2018; Fukue: Kanaya et al., 2016; Okinawa: Kunwar and Kawamura, 2014; Huaniao Island: Wang et al., 2014). Three CNEMC sites are marked in grey (Qingdao, Shanghai, and Fuzhou). Two research cruise tracks are represented by green line (Dongfanghong II from 17 March to 22 April 2014: Luo et al., 2016; Feng et al., 2017) and orange line (KEXUE-1 from 18 May to 12 June 2014: Kang et al., 2018), respectively. AERONET sites are represented by triangles with numbers (1-Ussuriysk, 2-Yonsei_University, 3-Gwangju_GIST, 4-EPA-NCU, 5-Chen-Kung_Univ, 6-Fukuoka). Full names of abbreviations are given in the text.



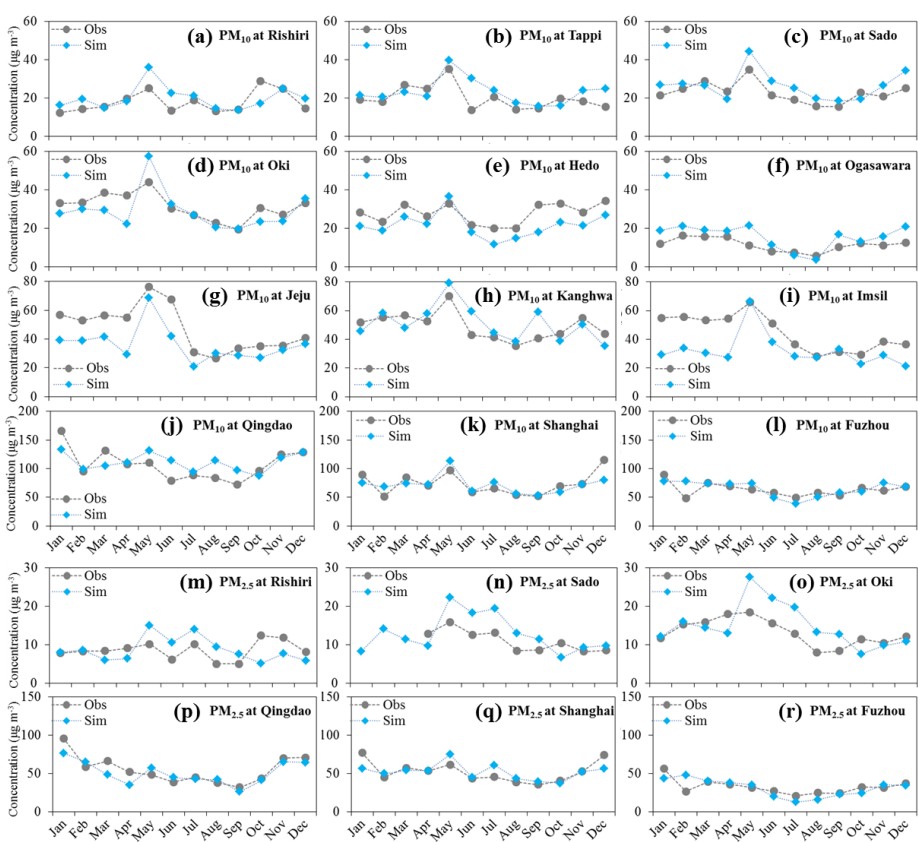

Figure 2. The model simulated (Sim) and observed (Obs) monthly $PM_{10}$ (a~l) and $PM_{2.5}$ (m~r) concentrations at EANET and CNEMC sites for the year 2014. The monthly data were averaged from hourly observations and the simulations were sampled according to the observations.

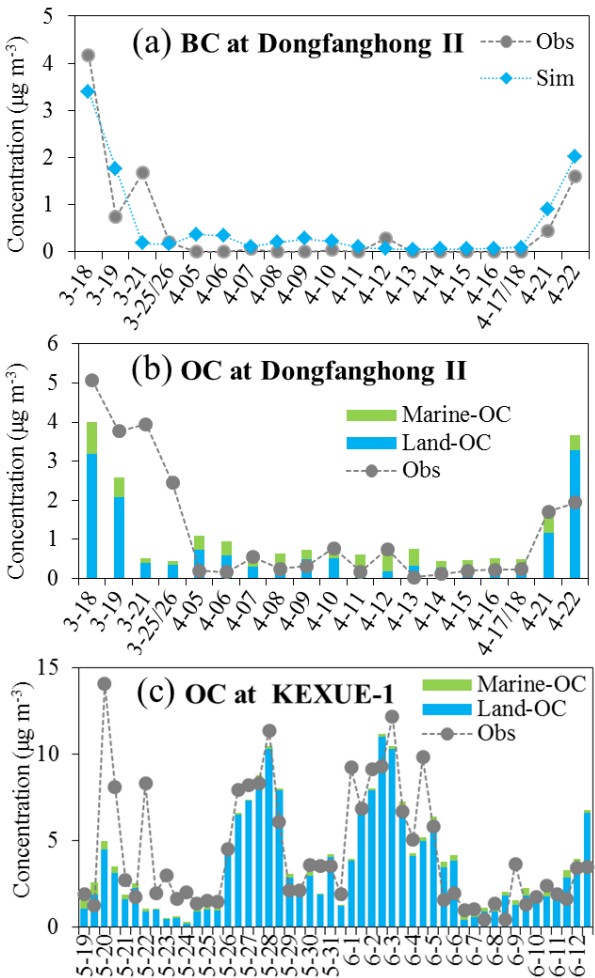

Figure 3. The model simulated (bars) and observed (dotted lines) daily BC and OC concentrations from the spring campaign (a, b) and half-day OC concentrations from the early summer campaign (c). The modeled total OC concentration was decomposed into those from marine (green bars) and land (blue bars) sources.

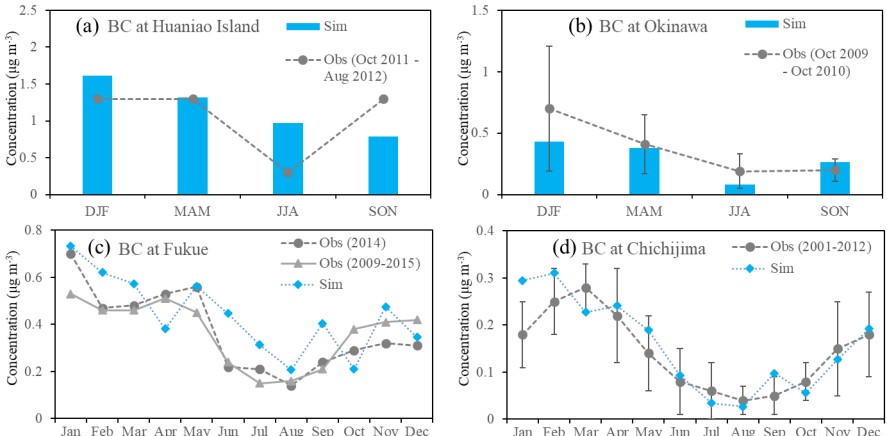

Figure 4. The model simulated (bars) and observed (dotted lines) BC concentrations at different sites. Seasonal mean concentrations were provided at (a) Huaniao Island (Wang et al., 2015) and (b) Okinawa (Kunwar and Kawamura, 2014) while monthly mean concentrations were provided at (c) Fukue (Kanaya et al., 2016) and (d) Chichijima Island (Boreddy et al., 2018). Standard deviations were available at Okinawa and Chichijima. The simulation is for the year 2014.

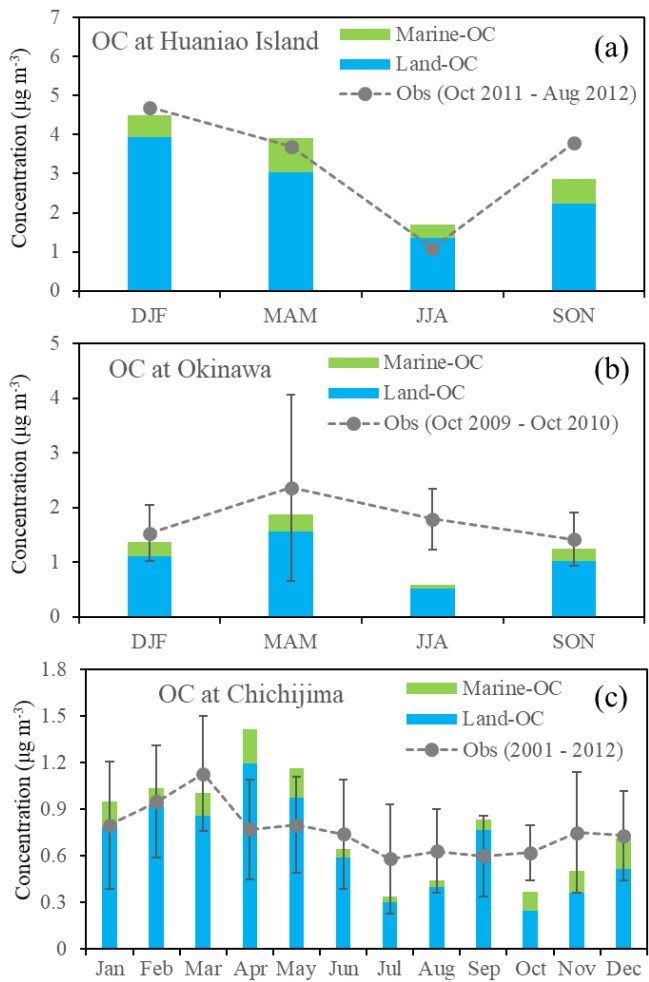

Figure 5. The model simulated (bars) and observed (dotted lines) OC concentrations at different sites. Seasonal mean concentrations were provided at (a) Huaniao Island (Wang et al., 2015) and (b) Okinawa (Kunwar and Kawamura, 2014) while monthly mean concentrations were provided at (c) Chichijima Island (Boreddy et al., 2018). Standard deviations were available at Okinawa and Chichijima. The modeled OC concentrations were decomposed to marine (green bars) and land (blue bars) sources. The simulation is for the year 2014.



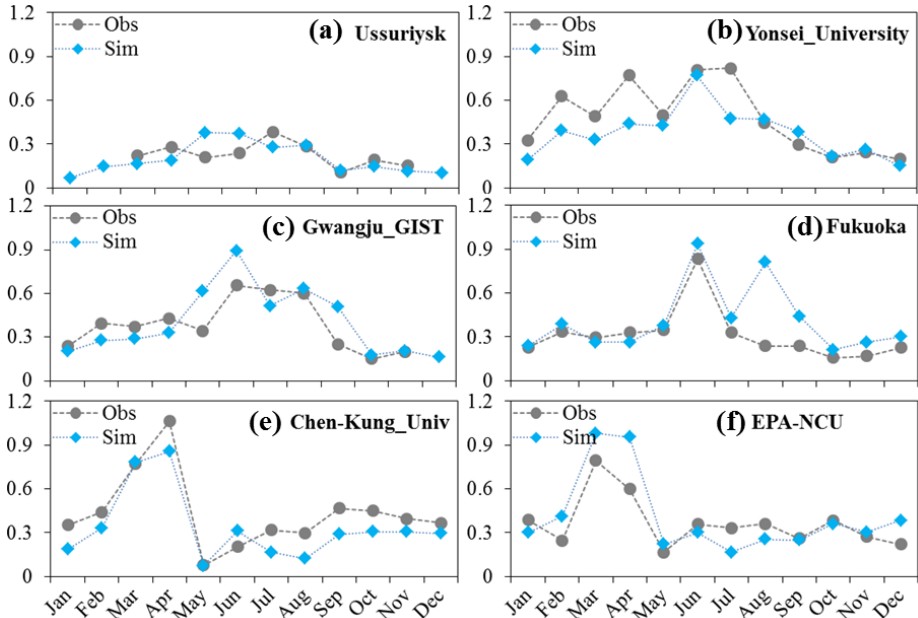

Figure 6. The model simulated (Sim) and observed (Obs) monthly mean AOD at 6 AERONET sites for
the year 2014. The monthly mean observations were calculated from hourly data and the corresponding
simulations were sampled according to the observations.

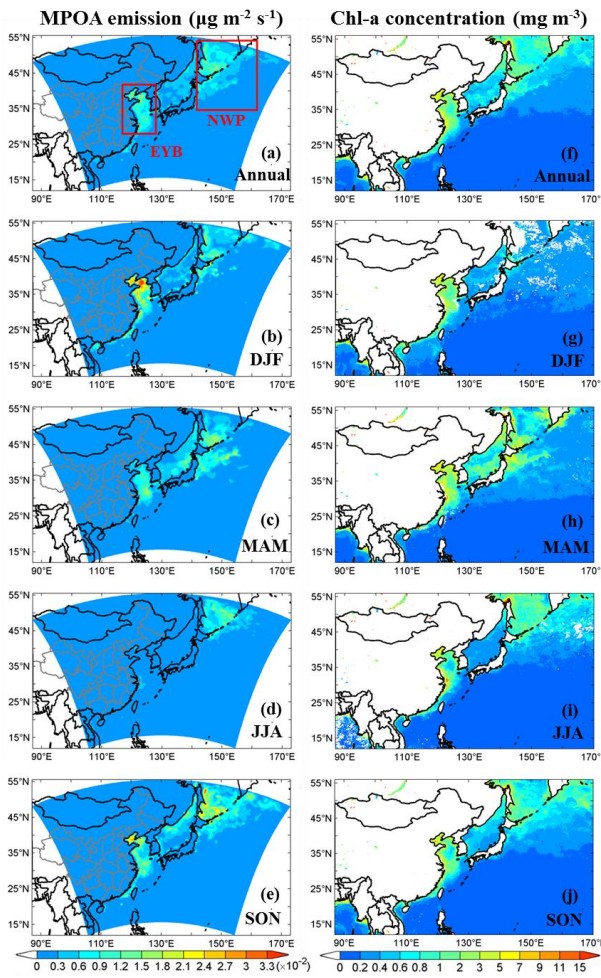

Figure 7. Model simulated annual and seasonal mean distributions of MPOA emissions (a~e) and VIIRS retrieved surface sea water chlorophyll-a (Chl-a) concentrations (f~j). Two hotspot regions are marked with red boxes: the region including the East China Sea, the Yellow Sea, and the Bohai Sea (EYB, 27~40°N, 115~123°E) and the region including the northern parts of the western Pacific to the northeast of Japan (NWP, 35~55°N, 140~160°E). Units are given in parentheses.

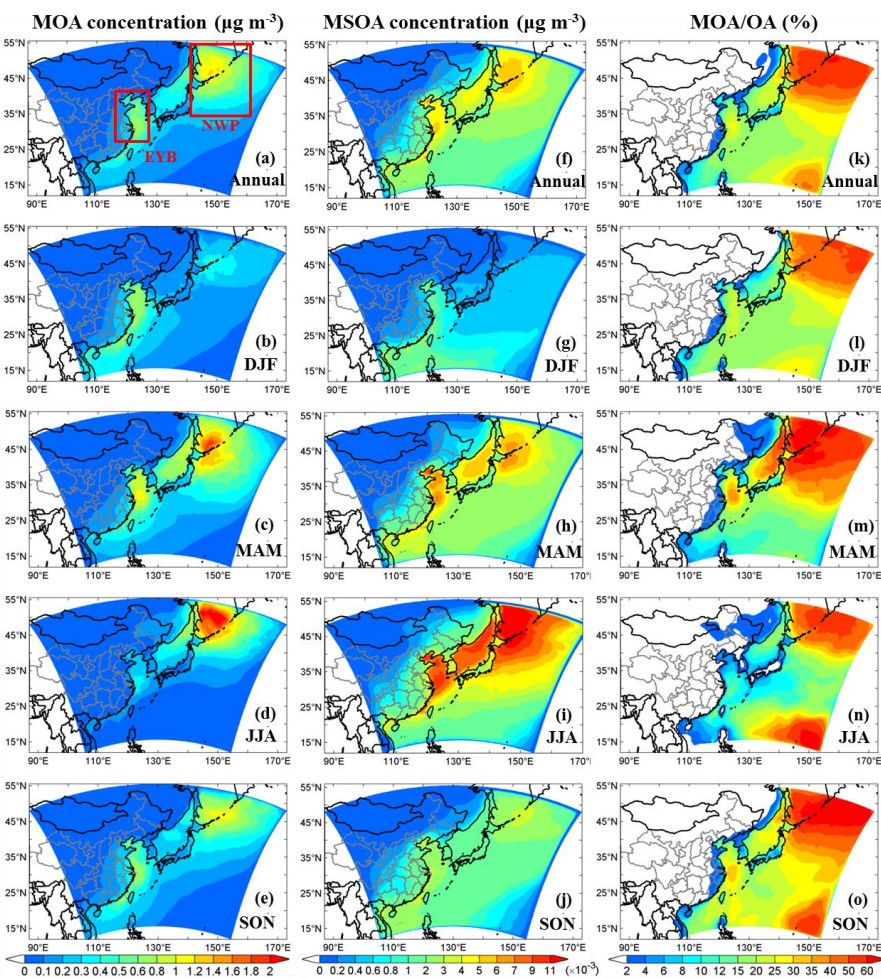

Figure 8. Model simulated annual and seasonal mean near surface MOA (primary+secondary) concentrations (a~e), near surface MSOA concentrations (f~j), and percentage contributions of MOA to total OA (k~o). The two regions of the EYB (27~40°N, 115~123°E) and the NWP (35~55°N, 140~160°E) are marked in 8a. Units are given in parentheses.

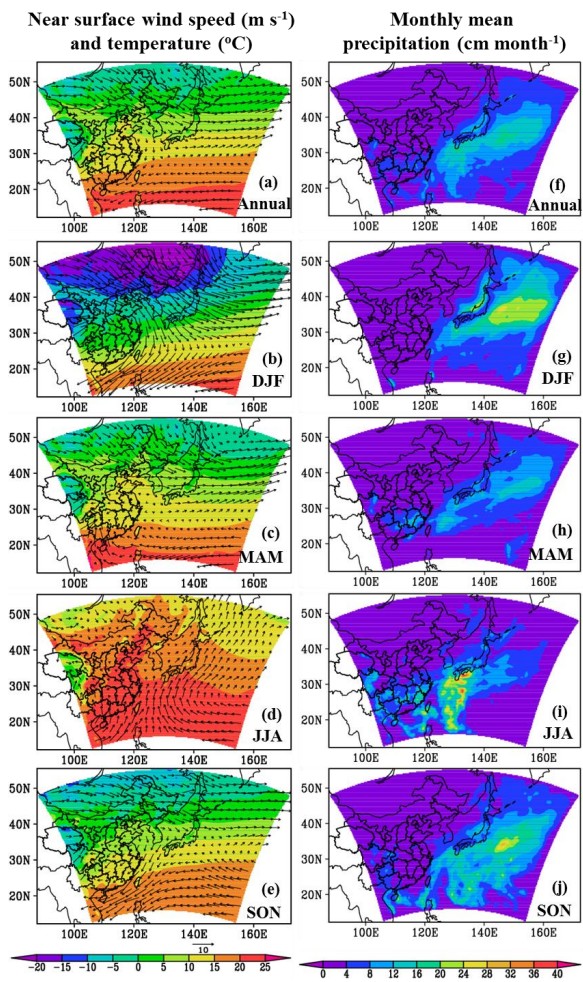

Figure 9. Model simulated annual and seasonal mean near surface temperatures (unite: °C) overlaid with wind vectors (unit: m s$^{-1}$) (a~e) and monthly mean precipitations (unit: cm month$^{-1}$) (f~j).

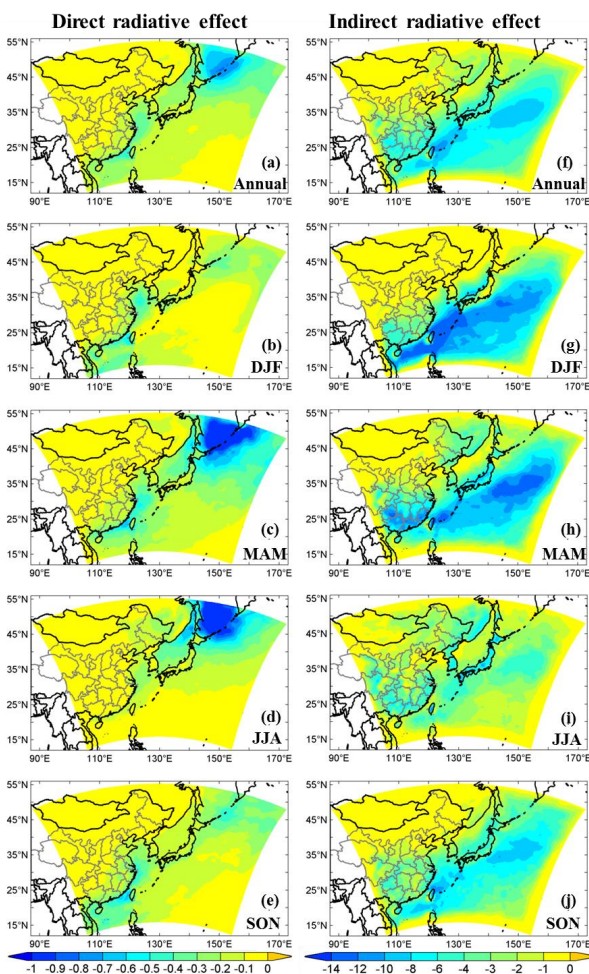

Figure 10. Model simulated annual and seasonal mean direct radiative effect due to MOA (DRE$_{MOA}$) (a~e) and indirect radiative effect due to MOA (IRE$_{MOA}$) (f~j) at the top of atmosphere (TOA) under all-sky condition (unit: W m$^{-2}$).


Table 1. Annual and seasonal performance statistics for hourly PM$_{10}$ and PM$_{2.5}$ concentrations (unit: μg m$^{-3}$) at EANET sites for the year 2014. Mean observation (Obs), mean simulation (Sim), correlation coefficient (R), and normalized mean bias (NMB in %) are listed. ANN=annual, DJF=December-January-February, MAM=March-April-May, JJA=June-July-August, and SON=September-October-November.

| Sites | Samples | ANN Obs | ANN Sim | ANN R | ANN NMB | DJF Obs | DJF Sim | DJF R | DJF NMB | MAM Obs | MAM Sim | MAM R | MAM NMB | JJA Obs | JJA Sim | JJA R | JJA NMB | SON Obs | SON Sim | SON R | SON NMB |
|---|---|---|---|---|---|---|---|---|---|---|---|---|---|---|---|---|---|---|---|---|---|
| **PM$_{10}$** | | | | | | | | | | | | | | | | | | | | | |
| Rishiri | 8381 | 18.0 | 19.9 | 0.53 | 11 | 13.6 | 18.4 | 0.65 | 35 | 20.1 | 23.0 | 0.56 | 15 | 15.2 | 19.4 | 0.42 | 28 | 23.0 | 18.8 | 0.51 | -18 |
| Tappi | 8584 | 20.1 | 23.2 | 0.49 | 15 | 17.4 | 22.3 | 0.54 | 28 | 29.0 | 28.0 | 0.59 | -4 | 16.1 | 23.8 | 0.18 | 48 | 17.6 | 18.5 | 0.39 | 5 |
| Sado | 8640 | 22.8 | 26.4 | 0.63 | 16 | 23.6 | 29.5 | 0.68 | 25 | 29.2 | 30.3 | 0.65 | 4 | 18.6 | 24.5 | 0.55 | 32 | 19.6 | 21.4 | 0.53 | 9 |
| Oki | 8424 | 31.3 | 29.2 | 0.68 | -7 | 33.2 | 31.1 | 0.65 | -7 | 40.2 | 37.5 | 0.71 | -7 | 26.7 | 26.6 | 0.61 | 0 | 25.8 | 22.3 | 0.66 | -14 |
| Hedo | 8008 | 27.7 | 21.7 | 0.56 | -22 | 28.8 | 22.4 | 0.66 | -22 | 30.5 | 28.3 | 0.58 | -7 | 20.7 | 14.8 | 0.54 | -28 | 30.9 | 20.8 | 0.34 | -33 |
| Ogasawara | 8120 | 11.5 | 15.7 | 0.48 | 36 | 13.4 | 20.3 | 0.38 | 52 | 14.2 | 19.7 | 0.40 | 39 | 7.0 | 6.8 | 0.46 | -2 | 11.2 | 15.0 | 0.30 | 34 |
| Jeju | 7101 | 46.9 | 36.9 | 0.64 | -21 | 50.1 | 38.2 | 0.71 | -24 | 62.6 | 46.6 | 0.66 | -26 | 36.4 | 31.5 | 0.44 | -13 | 34.7 | 29.3 | 0.44 | -15 |
| Kanghwa | 8524 | 49.2 | 51.2 | 0.59 | 4 | 50.2 | 46.0 | 0.60 | -8 | 59.9 | 61.9 | 0.66 | 3 | 40.0 | 47.2 | 0.47 | 18 | 46.5 | 49.3 | 0.38 | 6 |
| Imsil | 8383 | 44.5 | 32.3 | 0.58 | -27 | 48.8 | 27.9 | 0.63 | -43 | 58.0 | 42.1 | 0.62 | -27 | 38.4 | 31.1 | 0.47 | -19 | 33.0 | 28.2 | 0.42 | -15 |
| Average | 74165 | 30.0 | 28.5 | 0.65 | -5 | 30.8 | 28.3 | 0.67 | -8 | 37.9 | 35.2 | 0.65 | -7 | 23.9 | 25.1 | 0.59 | 5 | 26.9 | 25.0 | 0.58 | -7 |
| **PM$_{2.5}$** | | | | | | | | | | | | | | | | | | | | | |
| Rishiri | 8331 | 8.6 | 8.7 | 0.54 | 0 | | | | | | | | | | | | | | | | |
| Sado | 6517 | 11.0 | 13.4 | 0.53 | 21 | 8.1 | 7.4 | 0.78 | -8 | 9.2 | 9.2 | 0.56 | 0 | 7.2 | 11.5 | 0.54 | 59 | 10.0 | 6.7 | 0.31 | -33 |
| Oki | 8410 | 13.1 | 15.0 | 0.64 | 14 | 8.5 | 9.8 | 0.60 | 14 | 14.4 | 16.1 | 0.63 | 12 | 11.4 | 16.8 | 0.47 | 48 | 9.1 | 9.1 | 0.24 | 0 |
| Average | 23258 | 10.9 | 12.3 | 0.61 | 12 | 13.0 | 12.9 | 0.77 | -1 | 17.4 | 18.7 | 0.64 | 8 | 12.1 | 18.3 | 0.55 | 51 | 10.1 | 10.0 | 0.39 | -1 |



Table 2. Same as Table 1 but for CNEMC sites.

| Sites | Samples | ANN | | | | DJF | | | | MAM | | | | JJA | | | | SON | | | |
|---|---|---|---|---|---|---|---|---|---|---|---|---|---|---|---|---|---|---|---|---|---|
| | | Obs | Sim | R | NMB | Obs | Sim | R | NMB | Obs | Sim | R | NMB | Obs | Sim | R | NMB | Obs | Sim | R | NMB |
| **PM$_{10}$** | | | | | | | | | | | | | | | | | | | | | |
| Qingdao | 7622 | 107.0 | 108.6 | 0.61 | 1 | 131.0 | 124.3 | 0.76 | -5 | 117.3 | 109.9 | 0.49 | -6 | 83.6 | 108.4 | 0.64 | 30 | 97.1 | 101.4 | 0.59 | 4 |
| Shanghai | 7581 | 73.4 | 70.5 | 0.55 | -4 | 93.9 | 81.1 | 0.72 | -14 | 83.2 | 80.2 | 0.58 | -4 | 59.6 | 64.0 | 0.37 | 7 | 64.8 | 60.6 | 0.43 | -7 |
| Fuzhou | 7610 | 63.7 | 63.8 | 0.38 | 0 | 69.9 | 72.9 | 0.30 | 4 | 69.8 | 72.6 | 0.32 | 4 | 55.3 | 45.8 | 0.28 | -17 | 60.5 | 64.5 | 0.30 | 7 |
| Average | 22813 | 81.6 | 80.7 | 0.65 | -1 | 98.4 | 92.6 | 0.74 | -6 | 89.9 | 86.7 | 0.58 | -4 | 66.0 | 72.0 | 0.61 | 9 | 74.1 | 75.4 | 0.51 | 2 |
| **PM$_{2.5}$** | | | | | | | | | | | | | | | | | | | | | |
| Qingdao | 7627 | 55.2 | 48.7 | 0.72 | -12 | 75.1 | 67.8 | 0.83 | -10 | 56.3 | 43.7 | 0.61 | -22 | 40.5 | 42.8 | 0.60 | 6 | 48.2 | 43.9 | 0.74 | -9 |
| Shanghai | 7724 | 51.9 | 51.8 | 0.62 | 0 | 68.0 | 59.6 | 0.80 | -12 | 57.2 | 57.5 | 0.60 | 0 | 42.6 | 49.8 | 0.46 | 17 | 42.6 | 42.9 | 0.51 | 1 |
| Fuzhou | 7641 | 32.3 | 30.0 | 0.44 | -7 | 40.3 | 40.2 | 0.25 | 0 | 35.8 | 36.8 | 0.37 | 3 | 24.0 | 15.8 | 0.38 | -34 | 29.2 | 27.3 | 0.29 | -7 |
| Average | 22992 | 46.6 | 43.4 | 0.70 | -7 | 61.1 | 55.8 | 0.78 | -9 | 49.7 | 45.6 | 0.63 | -8 | 35.6 | 35.5 | 0.62 | 0 | 39.9 | 38.0 | 0.62 | -5 |





Table 3. Performance statistics for BC and OC from the two research campaigns in 2014. BC and OC were measured on Dongfanghong II during the spring campaign whereas only OC were collected on KEXUE-1 during the early summer campaign. Mean observation (Obs), mean simulation (Sim), correlation coefficient (R), and normalized mean bias (NMB in %) are listed. The modeled concentrations of marine-OC (including MPOA and MSOA) and its contribution to total OC were estimated.

| | Dongfanghong II | | | KEXUE-1 | |
|---|---|---|---|---|---|
| | BC | OC | Marine-OC (% in OC) | OC | Marine-OC (% in OC) |
| Samples | 19 | 19 | | 51 | |
| Obs (µg m⁻³) | 0.49 | 1.20 | | 4.26 | |
| Sim (µg m⁻³) | 0.55 | 1.14 | 0.33 (29%) | 3.68 | 0.23 (6%) |
| R | 0.87 | 0.66 | | 0.75 | |
| NMB (%) | 13 | -5 | | -13 | |

Table 4. Comparison of model simulated and observed seasonal BC and OC



concentrations (unit: μg m$^{-3}$) at Huaniao Island and Okinawa. The modeled concentrations of marine-OC and its contribution to total OC were estimated. ANN=annual, DJF=December-January-February, MAM=March-April-May, JJA=June-July-August, and SON=September-October-November.

| | | Time | ANN[c] | DJF | MAM | JJA | SON | Reference |
|---|---|---|---|---|---|---|---|---|
| **BC** | | | | | | | | |
| Huaniao Island[a] | Obs | Oct 2011~ Aug 2012 | 1.1 | 1.3 | 1.3 | 0.3 | 1.3 | Wang et al., 2015 |
| | Sim | 2014 | 1.2 | 1.6 | 1.3 | 1.0 | 0.8 | |
| Okinawa[b] | Obs | Oct 2009 ~ Oct 2010 | 0.38 | 0.70 | 0.41 | 0.19 | 0.20 | Kunwar and Kawamura, 2014 |
| | Sim | 2014 | 0.29 | 0.43 | 0.38 | 0.08 | 0.26 | |
| **OC** | | | | | | | | |
| Huaniao Island | Obs | Oct 2011~ Aug 2012 | 3.3 | 4.7 | 3.7 | 1.1 | 3.8 | Wang et al., 2015 |
| | Sim | 2014 | 3.2 | 4.5 | 3.9 | 1.7 | 2.9 | |
| | Marine-OC (% in OC) | | 0.6 (19%) | 0.56 (12%) | 0.88 (22%) | 0.32 (19%) | 0.65 (23%) | |
| Okinawa | Obs | Oct 2009~ Oct 2010 | 1.8 | 1.5 | 2.4 | 1.8 | 1.4 | Kunwar and Kawamura, 2014 |
| | Sim | 2014 | 1.3 | 1.4 | 1.9 | 0.6 | 1.2 | |
| | Marine-OC (% in OC) | | 0.21 (17%) | 0.25 (18%) | 0.32 (17%) | 0.06 (10%) | 0.23 (18%) | |

a: The location of Huaniao Island is 30.86°N, 122.67°E.

b: The location of Okinawa Island is 26.15°N, 128.03°E.

c: The annual means are averages of the four seasonal means.




Table 5. Comparison of model simulated and observed monthly mean BC and OC concentrations (unit: μg m$^{-3}$) at Fukue and Chichijima Island. Marine-OC concentration and its contribution to total OC at Chichijima were estimated.

| Month | BC at Fukue[a] | | | BC at Chichijima[b] | | OC at Chichijima[b] | | |
|---|---|---|---|---|---|---|---|---|
| | Obs (2014) | Obs (2009-2015) | Sim (2014) | Obs (2001-2012) | Sim (2014) | Obs (2001-2012) | Sim (2014) | Marine-OC (% in OC) |
| Jan | 0.70 | 0.53 | 0.73 | 0.18 | 0.29 | 0.80 | 0.95 | 0.17(18%) |
| Feb | 0.47 | 0.46 | 0.62 | 0.25 | 0.31 | 0.95 | 1.03 | 0.11(11%) |
| Mar | 0.48 | 0.46 | 0.57 | 0.28 | 0.23 | 1.13 | 1.01 | 0.15(15%) |
| Apr | 0.53 | 0.51 | 0.38 | 0.22 | 0.24 | 0.77 | 1.42 | 0.22(16%) |
| May | 0.56 | 0.45 | 0.56 | 0.14 | 0.19 | 0.80 | 1.17 | 0.19(16%) |
| Jun | 0.22 | 0.24 | 0.45 | 0.08 | 0.09 | 0.74 | 0.64 | 0.06 (9%) |
| Jul | 0.21 | 0.15 | 0.31 | 0.06 | 0.03 | 0.58 | 0.34 | 0.04(11%) |
| Aug | 0.14 | 0.16 | 0.21 | 0.04 | 0.03 | 0.63 | 0.44 | 0.04 (9%) |
| Sep | 0.24 | 0.21 | 0.40 | 0.05 | 0.10 | 0.60 | 0.84 | 0.07 (8%) |
| Oct | 0.29 | 0.38 | 0.21 | 0.08 | 0.06 | 0.62 | 0.37 | 0.12(33%) |
| Nov | 0.32 | 0.41 | 0.48 | 0.15 | 0.13 | 0.75 | 0.50 | 0.14(28%) |
| Dec | 0.31 | 0.42 | 0.35 | 0.18 | 0.19 | 0.73 | 0.71 | 0.19(27%) |
| Annaul | 0.37 | 0.37 | 0.44 | 0.14 | 0.16 | 0.76 | 0.78 | 0.13(16%) |

a: Data at Fukue were derived from Kanaya et al. (2016). The location of Fukue is 32.75°N, 128.68°E.

b: Data at Chichijima Island were derived from Boreddy et al. (2018). The location of Chichijima Island is 27.07°N, 142.22°E.



Table 6. Performance statistics for hourly AOD (unitless) at AERONET sites for the year 2014. Mean observation (Obs), mean simulation (Sim), correlation coefficient (R), and normalized mean bias (NMB in %) are listed. IDs are marked in Figure 1.

| ID | Site | Obs | Sim | R | NMB | Samples |
|---|---|---|---|---|---|---|
| 1 | Ussuriysk | 0.22 | 0.21 | 0.41 | -6 | 945 |
| 2 | Yonsei_University | 0.48 | 0.37 | 0.67 | -23 | 1629 |
| 3 | Gwangju_GIST | 0.33 | 0.36 | 0.53 | 7 | 900 |
| 4 | EPA-NCU | 0.38 | 0.39 | 0.43 | 4 | 685 |
| 5 | Chen-Kung_Univ | 0.49 | 0.37 | 0.60 | -25 | 657 |
| 6 | Fukuoka | 0.28 | 0.34 | 0.50 | 18 | 1144 |
|  | Average | 0.37 | 0.34 | 0.56 | -8 | 5960 |

Table 7. Modeled domain and annual/seasonal mean MPOA emission rates, surface sea water chlorophyll-a (Chl-a) concentrations, and sea salt emission fluxes over the western Pacific of East Asia (Mean), the region including the East China Sea, the Yellow Sea, and the Bohai Sea (EYB) and the region including northern parts of western Pacific to the northeast of Japan (NWP).

|  | MPOA emission ($\times 10^{-2}$ µg m$^{-2}$ s$^{-1}$) | | | | Chl-a concentration (mg m$^{-3}$) | | | Sea salt emission flux (µg m$^{-2}$ s$^{-1}$) | | |
|---|---|---|---|---|---|---|---|---|---|---|
|  | Mean[a] | Max[b] | EYB[c] | NWP[d] | Mean[a] | EYB[c] | NWP[d] | Mean[a] | EYB[c] | NWP[d] |
| ANN | 0.16 | 1.8 | 0.65 | 0.40 | 1.17 | 3.51 | 0.96 | 0.36 | 0.18 | 0.59 |
| DJF | 0.18 | 3.6 | 1.19 | 0.33 | 0.67 | 3.20 | 0.37 | 0.63 | 0.35 | 1.09 |
| MAM | 0.17 | 2.5 | 0.41 | 0.43 | 0.97 | 4.00 | 1.13 | 0.30 | 0.11 | 0.61 |
| JJA | 0.08 | 1.9 | 0.12 | 0.29 | 1.07 | 3.14 | 0.90 | 0.14 | 0.04 | 0.15 |
| SON | 0.20 | 3.5 | 0.88 | 0.54 | 1.10 | 2.90 | 0.90 | 0.38 | 0.24 | 0.53 |

a: Mean over oceanic areas.

b: Maximums over oceanic areas.

c: Ocean areas within 27~40°N, 115~123°E.

d: Ocean areas within 35~55°N, 140~160°E.



Table 8. Modeled domain and annual/seasonal mean near surface MOA concentrations, MSOA concentrations, and MOA to total OA ratios over the western Pacific of East Asia (Mean), the EYB region, and the NWP region.

| | MOA concentration (µg m$^{-3}$) | | | | MSOA concentration (×10$^{-3}$ µg m$^{-3}$) | | | | MOA/OA (%) | | | |
|---|---|---|---|---|---|---|---|---|---|---|---|---|
| | Mean[a] | Max[b] | EYB[c] | NWP[d] | Mean[a] | Max[b] | EYB[c] | NWP[d] | Mean[a] | Max[b] | EYB[c] | NWP[d] |
| ANN | 0.27 | 1.2 | 0.48 | 0.59 | 2.2 | 6.9 | 4.1 | 3.8 | 26% | 62% | 13% | 42% |
| DJF | 0.21 | 0.8 | 0.54 | 0.23 | 0.7 | 3.2 | 1.0 | 0.4 | 24% | 57% | 11% | 36% |
| MAM | 0.37 | 1.9 | 0.62 | 0.81 | 2.7 | 10.5 | 5.3 | 4.1 | 26% | 69% | 15% | 52% |
| JJA | 0.23 | 2.3 | 0.22 | 0.8 | 3.9 | 13.6 | 7.5 | 8.3 | 23% | 69% | 6% | 32% |
| SON | 0.26 | 1.3 | 0.52 | 0.52 | 1.5 | 4.2 | 2.6 | 2.2 | 32% | 73% | 18% | 48% |

a: Mean over oceanic areas.

b: Maximums over oceanic areas.

c: Ocean areas within 27~40°N, 115~123°E.

d: Ocean areas within 35~55°N, 140~160°E.

Table 9. Modeled domain and annual/seasonal mean near surface wind speed, temperature, precipitation, and relative humidity (RH) over the western Pacific of East Asia (Mean), the EYB region, and the NWP region.

| | Wind speed (m s$^{-1}$) | | | Temperature (°C) | | | Precipitation (cm grid$^{-1}$ month$^{-1}$) | | | RH (%) | | |
|---|---|---|---|---|---|---|---|---|---|---|---|---|
| | Mean[a] | EYB[b] | NWP[c] | Mean[a] | EYB[b] | NWP[c] | Mean[a] | EYB[b] | NWP[c] | Mean[a] | EYB[b] | NWP[c] |
| ANN | 4.3 | 2.9 | 4.0 | 19.2 | 15.1 | 8.5 | 6.1 | 2.7 | 8.0 | 78 | 73 | 83 |
| DJF | 6.4 | 4.5 | 6.9 | 14.0 | 4.5 | 1.0 | 7.0 | 1.8 | 12.4 | 75 | 67 | 77 |
| MAM | 3.8 | 2.0 | 3.7 | 16.9 | 13.4 | 5.1 | 4.3 | 2.1 | 7.0 | 79 | 75 | 84 |
| JJA | 3.0 | 1.9 | 2.5 | 24.0 | 23.2 | 15.8 | 5.1 | 3.5 | 3.7 | 83 | 80 | 94 |
| SON | 4.1 | 3.1 | 3.1 | 21.7 | 17.9 | 12.0 | 7.9 | 3.2 | 9.0 | 76 | 71 | 77 |

a: Mean over oceanic areas.

b: Ocean areas within 27~40°N, 115~123°E.

c: Ocean areas within 35~55°N, 140~160°E.





Table 10. Modeled domain and annual/seasonal mean all-sky TOA direct radiative effect (DRE) and indirect radiative effects (IRE) due to MOA and due to all aerosols over the western Pacific of East Asia, the EYB region, and the NWP region. The units are W m$^{-2}$.

| | MOA | | | | All aerosols | | | |
|---|---|---|---|---|---|---|---|---|
| | Mean[a] | Max[b] | EYB[c] | NWP[d] | Mean[a] | Max[b] | EYB[c] | NWP[d] |
| DRE | | | | | | | | |
| ANN | -0.21 | -0.86 | -0.24 | -0.41 | -17.9 | -33 | -24.6 | -17.9 |
| DJF | -0.14 | -0.59 | -0.25 | -0.16 | -15.2 | -30 | -15.3 | -9.5 |
| MAM | -0.31 | -1.64 | -0.24 | -0.68 | -21.6 | -42 | -26.9 | -19.7 |
| JJA | -0.20 | -1.31 | -0.17 | -0.58 | -19.2 | -44 | -32.9 | -27.8 |
| SON | -0.17 | -0.65 | -0.28 | -0.23 | -15.4 | -32 | -23.1 | -14.5 |
| IRE | | | | | | | | |
| ANN | -4.2 | -12.1 | -2.2 | -4.1 | -13.3 | -28.9 | -5.2 | -11.4 |
| DJF | -6.0 | -19.3 | -2.4 | -4.0 | -16.0 | -41.6 | -4.5 | -8.6 |
| MAM | -5.0 | -14.8 | -1.8 | -5.1 | -15.4 | -38.2 | -4.2 | -12.0 |
| JJA | -1.9 | -6.4 | -1.5 | -2.5 | -9.4 | -27.6 | -5.5 | -12.0 |
| SON | -3.9 | -12.0 | -2.9 | -4.9 | -12.5 | -28.6 | -6.4 | -12.9 |

a: Mean over oceanic areas.

b: Maximums over oceanic areas.

c: 27~40°N, 115~123°E.

d: 35~55°N, 140~160°E.