# Peer review of "Seasonal characteristics of emission, distribution and radiative effect of marine organic aerosols over the western Pacific Ocean: an analysis combining observations with regional modeling"

_Atmospheric Chemistry and Physics, 2020_

## Referee Comment (RC1) · Anonymous Referee #1 · 6 Dec 2020

In "Seasonal characteristics of emission, distribution and radiative effect of marine organic aerosols over the western Pacific Ocean: an analysis combining observations with regional modeling," Li et al. examined both primary (MPOA) and secondary (MSOA) marine organic aerosol in the western North Pacific Ocean using a regional chemistry/aerosol-climate model. Model simulated aerosol concentrations were validated against observations. Key MOA source regions and their seasonality in the western North Pacific were identified. MPOA was found to be much more important than MSOA, and significant indirect radiative effect was found.

[Figure]

The study is thorough and covers interesting findings. However, the presentation is overtly lengthy and should be condensed. Clarification on a few key details of the analysis should also be addressed, particularly as the lack of these pieces of information prevents interpretation of the results. I have trouble understanding how the direct and indirect radiative effects were calculated in this study. Detailed comments are provided below.

Major comments:

1. The paper would benefit from significant summarising and focusing. It may be better to summarise the model description (Sec. 2.1) in more concise terms (e.g. a summary list of all the processes included, and for the key processes, a sentence or so explaining how they are considered) and move the more detailed descriptions to the supplement. The same can also be said for the model validation (Sec. 3). While these are thoroughly done and should be documented, the shear length of the material in their entirety distracts from the storyline of the paper. Replacing with a concise summary in the main text and moving the details of the validation (with the many figures and tables) to the supplement would be helpful.

2. Section 2.3.1: please add how the emitted number is determined alongside emitted mass. The discussion on size distribution between lines 263-272 may fit better here, but additional information should also be given with regards to how these fit in the size bins.

3. L773-775: one cannot make an estimation of the East Asian contribution based on a mixture of regional and global model results, especially since just above, it's noted that the regional model produced higher emission rates than global models for the same region. Note also in the abstract (L30) and conclusion (L1080).

4. L801-803: as noted above, the potential difference due to different model and study time frame may be too large to draw any such conclusion about the ratio to global emission, even approximated. Was Arnold et al.'s West Pacific average similar to results from the current study?

5. L933-934: how much do the noMOE and FULL simulations differ aside from the added MOA? In terms of properties relevant to DRE: are there differences in the wet and dry deposition that impacted other aerosol species? If yes, how much do these account for the diagnosed DRE? Given that DRE is stated to be calculated under all-sky conditions, does this mean potential differences in cloud cover due to rain suppression by MOA could also play a role? Please specify exactly which variables are used for calculating the DRE_MOA (L933-934), beyond stating that it's a subtraction of the two simulations. Related to this, how was all-aerosol DRE calculated (Table 10)?

6. L977: how is IRE_MOA calculated? Also, the MSOA (L1044), sea salt and all aerosol (L1033-1043 and Table 10) IRE? I have trouble seeing how these can be properly determined from the same two simulations. Please specify the exact variables/equation/diagnostic procedure. Which processes and indirect radiative effects are captured by this definition?

7. Does cloud fraction differ between the two simulations due to rain suppression? Assuming that this is included in the calculated IRE_MOA, what proportion of IRE_MOA is related to changes in cloud microphysical properties (e.g. CDNC/Nc) and how much to changes in macroscopic properties (e.g. cloud cover, precipitation)? Could any of the changes in macroscopic properties be resulting from dynamical feedback? More information would be needed to properly interpret the high IRE_MOA despite relatively low MOA concentration (for instance, compared to sea salt) and to compare to other studies.

8. Given our limited understanding of many of MOA's properties, what's the (potential) sensitivity of the results to assumptions made in the model setup?

More minor comments:

9. A suggestion for consideration, since from my understanding, ACP does not limit

the abstract length: the abstract as it stands right now reads more like a conclusion/summary. Could it be pared down more? (for instance, what are the three key findings of this study?)

10. The title states "...an analysis combining observations with regional modeling," but as I understand it, the observations were not used to bias correct or calibrate the model in any way. In this sense the observations were used purely for model validation, and as such this wording is perhaps misleading (there's no "combining" involved). Instead, perhaps something more along the lines of "model validation and regional modeling" would be more accurate, if the authors decide to retain a focus on the model validation part.

11. L145-147: "supposed to" is not a good word choice here. If one wants to express uncertainty, "may" could be a good replacement. Reference(s) for this claim should also be added.

12. L166-168: the internally mixed anthropogenic aerosol is assumed to have a fixed distribution that does not change shape following activation and sedimentation? If yes, has there been studies justifying this choice? I wonder if the anthropogenic aerosol may be represented by size bins, as is done for the natural aerosols. If yes, please clarify.

13. L168: "Natural aerosols (mineral dust and sea salt)" and MPOA? Also, in general, how do MPOA and MSOA fit in these model descriptions? E.g. on L175: what's the hygroscopicity of MPOA and MSOA? The same as POA and SOA?

14. L210: OMss is the organic mass fraction of sea spray aerosol, not sea salt.

15. L220-221: please add reference for the OM/OC ratio

16. L262: please add reference/justification of choice for the MSOA soluble mass fraction

17. L282-283: perhaps I'm missing something, but if aerosol activation/Nc is already calculated based on the Abdul-Razzak and Ghan scheme, why is "the number of aerosols activated assumed to be equal to the number of aerosols scavenged in cloud"?

18. L359: Pearson correlation coefficient? Please be specific.

19. L718-719 and Figure 7: how does the sea salt emission look like? Perhaps add a column in figure 7 for sea salt emission (since EM_POA= $\alpha \times$Ess$\times$OMss)?

20. L756-759: please clarify if this is a speculation or confirmed by analysis. It's difficult to tell by the wording.

21. L777: "annual mean" as in "ng m-2 s-1" averaged over the area and over the whole year? Or "annual emission" in "Tg y-1"?

22. L788-790: what is the relevance of this sentence in the context of the current study? Does this then imply that it is reasonable to compare simulation to observation from a different year for order of magnitude check?

23. Figures 4 and 5: do the standard deviations represent variability of the monthly/seasonal mean across multiple years or also the variability within each month/season?

24. Figure 9: "mean monthly" (cm month-1; total monthly precipitation averaged over multiple months) instead of "monthly mean" (mm h-1; average precipitation rate over each month). Note both in figure title and caption.

25. Table 9: "cm grid-1 month-1": do the authors mean cm month-1?

---

## Referee Comment (RC2) · Anonymous Referee #2 · 7 Dec 2020

This study focuses on primary and secondary marine aerosols over the Western Pacific Ocean and their radiative effect. The findings are interesting and this study is well suited for ACP. In its present form, I found this study is too lengthy which makes it challenging to read. Important methodological information and analysis are missing regarding the radiative impact of MOA.

Abstract is very long. I recommend the authors try to highlight no more than 2-3 key points.

[Figure]

line 25. It reads like a new finding but as far as I understand this just describes the parameterization of Gantt et al. This should be clarified

line 70. The introduction focuses primarily on literature published prior to 2012. A lot has been done both in terms of observations (field and lab) and in terms of parameterization since that needs to be discussed by the authors.

Here are a couple of studies (by no mean an exhaustive list) that the authors may want to consider

Conte, L., Szopa, S., Aumont, O., Gros, V., & Bopp, L. (2020). Sources and sinks of isoprene in the global open ocean: Simulated patterns and emissions to the atmosphere. Journal of Geophysical Research: Oceans, 125, e2019JC015946. https://doi.org/10.1029/2019JC015946

Bates, T. S., Quinn, P. K., Coffman, D. J., Johnson, J. E., Upchurch, L., Saliba, G., et al. (2020). Variability in Marine Plankton Ecosystems Are Not Observed in Freshly Emitted Sea Spray Aerosol Over the North Atlantic Ocean. Geophysical Research Letters, 47, e2019GL085938. https://doi.org/10.1029/2019GL085938

Quinn, P. K., Bates, T. S., Coffman, D. J., Upchurch, L., Johnson, J. E., Moore, R., et al. (2019). Seasonal variations in western North Atlantic remote marine aerosol properties. Journal of Geophysical Research: Atmospheres, 2019; 124: 14240– 14261. https://doi.org/10.1029/2019JD031740

Brüggemann, M., Hayeck, N. & George, C. Interfacial photochemistry at the ocean surface is a global source of organic vapors and aerosols. Nat Commun 9, 2101 (2018). https://doi.org/10.1038/s41467-018-04528-7

Betram et al. Sea spray aerosol chemical composition: elemental and molecular mimics for laboratory studies of heterogeneous and multiphase reactions. Chemical Society Reviews, 31 Mar 2018, 47(7):2374-2400 DOI: 10.1039/c7cs00008a

Quinn, P., Coffman, D., Johnson, J. et al. Small fraction of marine cloud condensation nuclei made up of sea spray aerosol. Nature Geosci 10, 674–679 (2017). https://doi.org/10.1038/ngeo3003

S. M. Burrows and O. Ogunro and A. A. Frossard and L. M. Russell and P. J. Rasch and S. M. Elliott A physically based framework for modeling the organic fractionation of sea spray aerosol from bubble film Langmuir equilibria Atmospheric Chemistry and PhysicsÂă 14Âă 13601–13629Âă (2014) https://doi.org/10.5194/acp-14-13601-2014

line 169 It would be worth mentioning that this range exceeds the valid range for Monahan ($0.8 < r80 < 10mm$)

line 202 GEIA is a portal for many different inventories. Are the authors using a climatology of MEGANv2? That would be surprising since very detailed year-specific inventories are used for anthropogenic and biomass burning emissions. Please clarify.

line 226 It seems there could be a lot of other reasons for this difference. MODIS vs VIIRS Chl-A, differences in wind speed and sea salt parameterizations.

line 229 This section completely ignores the abiotic source of isoprene (see references above), which may be as large if not larger than the biological source in marine environments.

line 265 I may be missing something but I am not sure how to reconcile the source function for sea salt (radius > 0.1um) which is used to derive MOA emissions, with the assumed diameter of MOA (0.1 um). Please clarify.

line 267 do the authors also use 5 size bins to represent MPOA or do they only consider sub-micron MPOA for this work?

Section 3. I suggest to have a section devoted solely to describing the different sources of observations, such that section 3 can focus solely on the model performance. The analysis of BC should focus more clearly on how these observations can help understand marine organic aerosols. The detailed BC analysis presented here could be moved to supporting materials, which would help shorten the manuscript.

It would be helpful to evaluate the simulated Na+ with the UMiami/Prospero dataset.

line 385-396. Suggests removing or moving to supporting materials

line 471 The model does not seem to capture the variability in OA from 4/7 to 4/13 (e.g., it shows high values in 4/11 for instance). Could the authors comment on this discrepancy? Could the model underestimate land SOA (which will not correlate with BC) over this time period?

linee 478 This needs some reference. What is the size range of fungi spores?

line 583. Many aspects of the overall OA budget remain challenging to represent (https://acp.copernicus.org/articles/20/2637/2020/). The contributions of MOA is fairly small at most sites. Could optimization of the land source of SOA or the removal of OA also reduce the model bias?

line 626. Does MOZART include MOA emissions?

line 711. I suggest to also compare with satellite AOD (MODIS, MISR, VIIRS) so that performances over the Western Pacific Ocean can be better assessed.

lines 771-775. You cannot mix your regional estimate with previous global estimate. Instead you would need to run your model globally to draw such conclusion. This also means that the abstract and conclusion need to be revised. Same issue on line 801.

line 885. Please clarify why this is impressive.

Sections 4.3 and 4.4

While there is an excess of details in previous sections, more analysis/method descriptions are needed here.

Please provide the equations to estimate DRE and IDRE. It seems that you would need more than 2 experiments to estimate the IDRE for the different types of aerosols. Are these estimates based on an ensemble of 1 yr simulations? Are the differences

shown here significant (relative to natural variability)? In general the authors need to better quantify the uncertainties associated with their estimates? This is especially important for the IRE_MOA. The authors also need to discuss their findings in the context of recent work that suggests a small role of SSA for CCNs (e.g., Quinn et al. DOI: 10.1038/NGEO3003)

―――――――――――――――――――

---

## Author Comment (AC1) · 21 Jan 2021

**Responses to the reviewer's comments**

**MS No.: acp-2020-1016**

Title: Seasonal characteristics of emission, distribution and radiative effect of marine organic aerosols over the western Pacific Ocean: an analysis combining observations with regional modeling

The authors greatly appreciate the valuable and constructive comments from the two reviewers, which have helped us improve the manuscript. We have addressed their comments carefully and revised the manuscript accordingly by taken their good suggestions into account. The detailed responses (blue font) are as follows:

**Response to Referee #1**

**General comments:**

In "Seasonal characteristics of emission, distribution and radiative effect of marine organic aerosols over the western Pacific Ocean: an analysis combining observations with regional modeling," Li et al. examined both primary (MPOA) and secondary (MSOA) marine organic aerosol in the western North Pacific Ocean using a regional chemistry/aerosol-climate model. Model simulated aerosol concentrations were validated against observations. Key MOA source regions and their seasonality in the western North Pacific were identified. MPOA was found to be much more important than MSOA, and significant indirect radiative effect was found. The study is thorough and covers interesting findings. However, the presentation is overtly lengthy and should be condensed. Clarification on a few key details of the analysis should also be addressed, particularly as the lack of these pieces of information prevents interpretation of the results. I have trouble understanding how the direct and indirect radiative effects were calculated in this study. Detailed comments are provided below.

**Reply:** Thanks for the valuable and constructive comments which help us improve the manuscript. We have responded to your comments in detail and revised the manuscript as your suggestions.

**Major comments:**

1. The paper would benefit from significant summarising and focusing. It may be better to summarise the model description (Sec. 2.1) in more concise terms (e.g. a summary list of all the processes included, and for the key processes, a sentence or so explaining how they are considered) and move the more detailed descriptions to the supplement. The same can also be said for the model validation (Sec. 3). While these are thoroughly done and should be documented, the shear length of the material in their entirety distracts from the storyline of the paper. Replacing with a concise summary in the main text and moving the details of the validation (with the many figures and tables) to the supplement would be helpful.

Reply: Thank you very much for your good suggestions. We revise the model description by reorganizing the sections and by moving some detailed descriptions (including formulas) to the supplement. We also reduce the length of section 3 by summarizing the model validation,

deleting detailed descriptions on validation for black carbon and gas precursors and moving relevant figures to the supplement (also suggested by the second reviewer).

2. Section 2.3.1: please add how the emitted number is determined alongside emitted mass. The discussion on size distribution between lines 263-272 may fit better here, but additional information should also be given with regards to how these fit in the size bins.

Reply: All aerosols species are assumed to have a log-normal distribution, the number concentration of aerosol species is calculated by mass concentration according to the following equation (Curci et al., 2015):

$$N_i = \frac{M_i}{\frac{4}{3}\pi r_i^3 \rho_i exp\left(\frac{9}{2}log^2\sigma_i\right)}$$

Where  $r_i$  is the geometric mean radius of aerosol species i,  $\sigma_i$  is the geometric standard deviation,  $\rho_i$  is the density of aerosol species. The  $r_i$  and  $\sigma_i$  of MPOA are derived from the cruise measurements over the western Pacific Ocean. We add the sentence "The number concentration is calculated by mass concentration as the formula in Curci et al. (2015)" in section 2.2 in the revised version. MOA (includes MPOA and MSOA) with size larger than 0.1 µm is not considered because they only account for a very small fraction of sea spray aerosol according to cruise measurements in the western Pacific Ocean (Feng et al., 2017).

**Reference**

Curci, G., Hogrefe, C., Bianconi, R., Im, U., Balzarini, A., Baró, R., Brunner, D., Forkel, R., Giordano, L., Hirtl, M., Honzak, L., Jiménez-Guerrero, P., Knote, C., Langer, M., Makar, P. A., Pirovano, G., Pérez, J. L., San José, R., Syrakov, D., Tuccella, P., Werhahn, J., Wolke, R., Žabkar, R., Zhang, J., and Galmarini, S.: Uncertainties of simulated aerosol optical properties induced by assumptions on aerosol physical and chemical properties: An AQMEII-2 perspective, Atmos. Environ., 115, 541–552, 2015.

3. L773-775: one cannot make an estimation of the East Asian contribution based on a mixture of regional and global model results, especially since just above, it's noted that the regional model produced higher emission rates than global models for the same region. Note also in the abstract (L30) and conclusion (L1080).

Reply: Yes, we delete relevant comparison and discussion throughout the manuscript.

4. L801-803: as noted above, the potential difference due to different model and study time frame may be too large to draw any such conclusion about the ratio to global emission, even approximated. Was Arnold et al.'s West Pacific average similar to results from the current study?

Reply: Yes, same as the above question, we delete such comparison between regional and global estimates. Arnold et al. (2009) used two methods to estimate marine isoprene emission flux, (1) a "bottom-up" scheme using satellite products and phytoplankton-specific isoprene productivity data; (2) a "top-down" scheme by minimizing the mean bias between the model and isoprene observations in the marine atmosphere remote from the continents. Their "bottom-up" scheme is similar to the parameterization used in this study. In general, our

modeled surface atmospheric isoprene concentrations were close to their "bottom-up" results (they didn't present global isoprene emission distribution). In the Figure 2b of Arnold et al. (2009), the annual mean surface atmospheric isoprene concentrations were 0.3~5 pptv over remote West Pacific (about 25~50°N and 130~180°E), correspondingly, our model results are approximately 0.1~4 pptv over the similar region.

5. L933-934: how much do the noMOE and FULL simulations differ aside from the added MOA? In terms of properties relevant to DRE: are there differences in the wet and dry deposition that impacted other aerosol species? If yes, how much do these account for the diagnosed DRE? Given that DRE is stated to be calculated under all-sky conditions, does this mean potential differences in cloud cover due to rain suppression by MOA could also play a role? Please specify exactly which variables are used for calculating the DRE\_MOA (L933-934), beyond stating that it's a subtraction of the two simulations. Related to this, how was all-aerosol DRE calculated (Table 10)?

**Reply:** We are sorry for the confusion. The direct radiative effect (DRE) is defined as the difference in net shortwave radiation flux at TOA (or at the surface) induced by aerosols (either individual aerosols or all aerosols, e.g., MPOA, MSOA, sea salt etc.) between cases with and without aerosols. The DRE in this study is derived by two calculations with and without aerosols (call two times in the radiation module at each time step, one with and one without aerosols) in one simulation. So, DRE reflects an instantaneous change in solar radiation fluxes induced by aerosols, without feedbacks from dry and wet depositions at this time step. The all-aerosol DRE is calculated using the same method, i.e., in the radiation module, at each time step, radiative fluxes are calculated twice with and without all aerosols and then DRE is derived from the difference between the two calculations, here all aerosols include anthropogenic aerosols internally mixed with each other and externally mixed with mineral dust and marine aerosols (sea salt, MPOA, MSOA).

The aerosol optical parameters (including extinction coefficient, single scattering albedo, and asymmetry factor) used to calculate DRE in the radiation module are derived from a Mie theory-based scheme developed by Ghan and Zaveri (2007), in which the aerosol optical parameters are pre-calculated by the Mie theory and then fitted by Chebyshev polynomials with a table of polynomial coefficients. The effect of water uptake is treated by the  $\kappa$ -Köhler parameterization, which calculates aerosol wet diameter due to hygroscopic growth. The bulk  $\kappa$  for internal mixture of aerosols is derived by the volume-weighted average of  $\kappa$  of each aerosol component, and the refractive index of internally mixed aerosols is calculated using the Maxwell-Garnett mixing rule. After obtaining the wet diameter and refractive index of the internally mixed aerosols (or a specific aerosol component), the aerosol optical properties can be derived from the Chebyshev fitting coefficients table. The advantage of this scheme is the much faster computational speed than traditional Mie calculation, with a similar level of accuracy. A more detailed description on parameters, method with formulas and procedure for calculating aerosol optical properties for DRE estimation is presented in a recent paper of ours (Li et al., 2020). To avoid repetition and for brevity, we briefly introduce the above method and cite this paper in the revised version.

DRE under all-sky conditions takes into account the cloud effect on clear-sky DRE (not the aerosol indirect effect on cloud nucleation and cloud properties), e.g., cloud layer at different altitude relative to aerosol layer affects aerosol reflectivity and DRE, the DRE of scattering aerosol under all-sky condition is smaller than that under clear-sky condition (Liao and Seinfeld, 1998). So, the change in cloud cover due to rain suppression by MOA does not affect DRE estimation, the aerosol's effect on cloud and potential feedback effects are considered in the IRE calculation which is described in detail as below.

**Reference**

Li Jiawei, Han Zhiwei, Wu Yunfei, Xiong Zhe, Xia Xiangao, Li Jie, Liang Lin, Zhang Renjian: Aerosol radiative effects and feedbacks on boundary layer meteorology and PM2.5 chemical components during winter haze events over the Beijing-Tianjin-Hebei region. Atmos. Chem. Phys., 20, 8659–8690, 2020.

Liao, H., and J. H. Seinfeld: Effects of clouds on direct aerosol radiative forcing of climate, J. Geophys. Res., 103, 3781–3788, 1998.

6. L977: how is IRE\_MOA calculated? Also, the MSOA (L1044), sea salt and all aerosol (L1033-1043 and Table 10) IRE? I have trouble seeing how these can be properly determined from the same two simulations. Please specify the exact variables/equation/diagnostic procedure. Which processes and indirect radiative effects are captured by this definition?

**Reply:** We are sorry for missing the description on IRE\_MOA calculation. As approaches in previous studies (Lohmann and Feichter, 2005; Wang and Penner, 2009; Leibensperger et al., 2012; Zhao et al., 2017), the indirect effect is defined as the difference in net shortwave radiation flux at TOA (or at the surface) induced by aerosols (either individual aerosols or all aerosols, e.g., MPOA, sea salt etc.) between cases with and without aerosols (or pre-industrial clean condition). The IRE due to individual or all aerosols are calculated through a series of simulations described as below:

$$\begin{split} & IRE_{mpoa} = (F \downarrow - F \uparrow)_{with mpoa} - (F \downarrow - F \uparrow)_{without mpoa} \\ & IRE_{msoa} = (F \downarrow - F \uparrow)_{with msoa} - (F \downarrow - F \uparrow)_{without msoa} \\ & IRE_{sea salt} = (F \downarrow - F \uparrow)_{with sea salt} - (F \downarrow - F \uparrow)_{without sea salt} \\ & IRE_{all} = (F \downarrow - F \uparrow)_{with all aerosols} - (F \downarrow - F \uparrow)_{without all aerosols} \end{split}$$

Where  $F\downarrow$  and  $F\uparrow$  are the incoming and outgoing shortwave radiation fluxes, respectively. Here all aerosols include both natural and anthropogenic aerosols.

The indirect effect consists of the first and second indirect effect. The first indirect effect is derived from the difference in the net shortwave radiation fluxes between two calculations (call two times, one with cloud optical parameters, e.g., cloud optical depth under background condition of cloud droplet number concentration ( $N_c$ ) and one with cloud optical parameters under  $N_c$  condition perturbed by aerosols, e.g., MPOA) at each time step in the radiation module, which reflects an instantaneous change in shortwave radiative flux due to aerosol perturbation to cloud properties ( $N_c$ , cloud effective radius and albedo etc.). As the commonly used method, the background condition (a pristine environment) is represented by prescribing a low bound of  $N_c$  of 10/cm3, which generally represent liquid stratiform cloud in clean marine conditions according to satellite observations and global model simulations (Bennartz, 2007; Hoose et al., 2009; Zeng et al., 2014; Zhao et al., 2017).

Details on calculation procedure for cloud properties and IRE are as follows:

While a specific aerosol component is considered or added (e.g. MPOA), Nc due to aerosol activation is diagnosed by the A-G scheme based on Köhler theory, then the cloud effective radius re is calculated as a function of Nc and cloud liquid water content following the approach of Martin et al. (1994), and the cloud optical properties (liquid cloud extinction optical depth, single scatter albedo, asymmetry factor etc.) are calculated by the scheme of Slingo et al. (1989), finally, shortwave radiation fluxes are calculated by the CCM3 radiation scheme (Kiehl et al., 1996), and the first indirect effect is derived from the difference in the net shortwave radiation fluxes between the two calculations (mentioned above) every time step within one simulation. The changes in the above cloud microphysical properties subsequently affect the conversion of cloud water to rainwater, which is a function of Nc (diagnosed above) and cloud liquid water content represented by the scheme of Beheng (1994) and further affect cloud properties, radiative fluxes and precipitation (namely the second indirect effect), and affect the first indirect effect in the next time step, so the second indirect effect influences the first indirect effect calculation through modifying cloud microphysical and optical properties (in part through altering precipitation and wet scavenging of aerosols and CCNs).

In summary, the indirect radiative effect (IRE) in this study represents processes through which aerosols perturb cloud microphysical and optical properties and solar radiation flux, that is an addition of aerosol components (e.g. MPOA) leads to increases in CCN and  $N_c$ , a decrease in  $r_e$ , and increases in cloud optical depth and cloud albedo (the first indirect effect), and leads to decreases in conversion rate from cloud water to rain water, increases in cloud water content, cloud amount, cloud optical depth and decreases in precipitation, which may in turn affect aqueous chemistry and scavenging of airborne aerosol loading and thus CCN and  $N_c$ . The first indirect effect results in a negative solar radiative effect at TOA due to increased cloud albedo and outgoing solar radiation, while the second indirect effect strengthens the negative effect.

**References**

- Bennartz, R.: Global assessment of marine boundary layer cloud droplet number concentration from satellite, J. Geophys. Res.,112, D02201, doi:10.1029/2006jd007547, 2007.
- Kiehl, J.T., Hack, J.J., Bonan, G.B., Boville, B.A., Briegleb, B.P., Williamson, D.L., Rasch, P.J.: Description of the NCAR Community Climate Model (CCM3), NCAR Technical Note, NCAR/TN-420+STR, p.152, 1996.
- Hoose C., Kristja'nsson J. E., Iversen T., A. Kirkevag, Ø. Seland, and Gettelman A.: Constraining cloud droplet number concentration in GCMs suppresses the aerosol indirect effect, Geophys. Res. Lett., 36, L12807, doi:10.1029/2009GL038568, 2009.
- Leibensperger E. M., Mickley L. J., Jacob D. J., Chen W.-T., Seinfeld J. H., Nenes A., Adams P. J., Streets D. G., Kumar N., and Rind D.: Climatic effects of 1950–2050 changes in US anthropogenic aerosols – Part 1: Aerosol trends and radiative forcing, Atmos. Chem. Phys., 12, 3333–3348, 2012.
- Lohmann, U. and Feichter J.: Global indirect aerosol effects: a review, Atmos. Chem. Phys., 5,

715-737, 2005.

- Martin, G. M., Johnson, D. W., and Spice, A.: The Measurements and Parameterization of Effective Radius of Droplets in Warm Stratocumulus Clouds, J. Atmos. Sci., 51, 1823–1842, 1994.
- Slingo A.: A GCM Parameterization for the Shortwave Radiative Properties of Water Clouds, J. Atmos. Sci., 46(10), 1419-1427, 1989.
- Wang, M., and Penner J. E.: Aerosol indirect forcing in a global model with particle nucleation, Atmos. Chem. Phys., 9, 239–260, 2009.
- Zeng, S., Riedi, J., Trepte, C. R., Winker, D. M. & Hu, Y. X.: Study of global cloud droplet number concentration with A-Train satellites. Atmos. Chem. Phys. 14, 7125–7134, 2014.
- Zhao, B., Liou, K. N., Gu, Y., Li, Q., Jiang, J. H., Su, H., He, C., Tseng, H. R., Wang, S., Liu, R., Qi, L., Lee, W. L., and Hao, J.: Enhanced PM2:5 pollution in China due to aerosol-cloud interactions, Sci. Rep., 7, 4453, https://doi.org/10.1038/s41598-017-04096-8, 2017.

7. Does cloud fraction differ between the two simulations due to rain suppression? Assuming that this is included in the calculated IRE\_MOA, what proportion of IRE\_MOA is related to changes in cloud microphysical properties (e.g. CDNC/Nc) and how much to changes in macroscopic properties (e.g. cloud cover, precipitation)? Could any of the changes in macroscopic properties be resulting from dynamical feedback? More information would be needed to properly interpret the high IRE\_MOA despite relatively low MOA concentration (for instance, compared to sea salt) and to compare to other studies.

**Reply:** As we discussed above, the indirect effect includes some feedback effects by the second indirect effect. To distinguish the IRE\_MOA between the first indirect effect related to changes in cloud microphysical properties (e.g. CDNC/Nc) and the second indirect effect related to macroscopic properties (e.g. cloud cover, precipitation), we conduct an additional sensitivity simulation from the base case by inactivating the second indirect effect (by assigning  $N_c$  to be the background 10/cm3 in the Beheng scheme) (i.e., just considering the first indirect effect). The contrast between the base case and the sensitivity case (Figure S5a,d) shows that the first indirect effect account for majority of the total indirect effect and the second indirect effect, which involves radiative, dynamic and precipitation feedbacks reinforces the first indirect effect (a stronger negative IRE in Figure S5a), because the cloud albedo is further enhanced resulting from increased cloud water content (together with decreased precipitation) due to weakened cloud water to rain water conversion, as clearly shown in Figure S5b,e over the western Pacific. These feedback processes are highly complex and nonlinear and beyond the scope of this manuscript, so we present a brief discussion about the relative roles of the first and second indirect effect, with the sensitivity model results and figure added in the supplement in the revised version.

The main reasons for the high IRE\_MOA in this study (relevant to sea salt) could be: 1.) higher number concentration of MOA than that of sea salt (see the figure below), because the geometric mean radius of MOA ( $0.05\mu$ m based on cruise measurement from Feng et al., 2017) is smaller than that of the fine mode sea salt ( $0.1-1.0\mu$ m); 2.) we assume a slight solubility of MOA with smaller molecule weight in this study, which could result in lower critical supersaturation for aerosol activation and more CCN. To address the uncertainty in

IRE\_MOA, we conduct additional simulations regarding MOA properties. We add the sensitivity simulations, discussions and comparison between our model results and other studies, e.g., Quinn et al. (2017) in the revised version (please see more details about the sensitivity simulations in response to the question 8 below and about the comparison with Quinn et al., 2017 in the response to the second reviewer).

Model simulated annual mean near surface aerosol number concentrations for (a) MOA and (b) sea salt (units:  $\#/cm^3$ ).

**References**

Feng, L.M., Shen, H.Q., Zhu, Y.J., Gao, H.W., and Yao, X.H.: Insight into Generation and Evolution of Sea-Salt Aerosols from Field Measurements in Diversified Marine and Coastal Atmospheres, Sci. Rep., 7, 41260; doi: 10.1038/srep41260, 2017.

8. Given our limited understanding of many of MOA's properties, what's the (potential) sensitivity of the results to assumptions made in the model setup?

Reply: Thanks for raising this important question which is also our interest. Yes, our current knowledge on the physical and chemical properties of MOA is still very limited, especially over the western Pacific Ocean of east Asia, although a few cruise measurements were carried out and some knowledge on MOA properties were gained. To address this uncertainty, three additional sensitivity simulations regarding MOA properties (note we focus on MPOA due to its dominant fraction in MOA as described in the manuscript) regarding particle size, solubility and molecule weight, which is crucial to aerosol activation are conducted for the entire year to provide a range of IRE due to MOA. These sensitivity experiments, discussions, relevant figures/tables are added to the section 4.4 in the revised version as follows:

"Due to our limited knowledge on MOA properties, there cloud be uncertainties in the estimated IREMOA. To address such uncertainty, three additional sensitivity simulations from the base case (results shown in Figure S6 and Table S5 in the revised version) were carried out regarding particle size, solubility and molecule weight, which are crucial to aerosol activation (note we focus on MPOA due to its dominant fraction in MOA as shown above). The first sensitivity simulation (SENS1) assumes a smaller geometric mean radius ( $0.03\mu m$  instead of  $0.05\mu m$  in the base case) for MPOA, resulting in a weaker domain-annual mean IREMOA (-3.5Wm-2) than that in the base case (-4.2 Wm-2) over the oceanic region (Figure S6b,

Table S5). The second sensitivity simulation (SENS2) assigns a lower solubility (0.03) with relatively large molecule weight (146 g mol-1) for MPOA (which is similar to the properties of adipic acid, Huff Hartz et al., 2006; Miyazaki et al., 2010) instead of the slight solubility (0.1) with a smaller molecule weight (90 g mol-1) (which is similar to the properties of oxalic acid, Roelofs, 2008; Miyazaki et al., 2010) in the base case, in this case, the IREMOA reduces to -2.8 Wm-2 (Figure S6c, Table S5). The third simulation (SENS3) combines the above two cases, assuming a smaller geometric mean radius as in SENS1 together with the lower solubility and larger molecule weight as in SENS2, it produces a further reduced IREMOA of -2.2 Wm-2 (Figure S6d, Table S5). The above sensitivity simulations exhibit a high sensitivity of IREMOA to the MPOA properties, and IREMOA accounts for approximately 28%, 22% and 17% of the total IRE by all aerosols in the three cases, respectively (note the total IRE also changes due to the changes in IREMOA in the sensitivity simulations), in contrast to the percentage contribution of 32% in the base case".

**References**

Huff Hartz Kara E., Tischuk Joshua E., Chan Man Nin, Chan Chak K., Donahue Neil M., Pandis Spyros N.: Cloud condensation nuclei activation of limited solubility organic aerosol, Atmospheric Environment 40, 605–617, 2006.

Miyazaki Yuzo, Kawamura Kimitaka, and Sawano Maki: Size distributions and chemical characterization of water-soluble organic aerosols over the western North Pacific in summer, J. Geophys. Res., 115, D23210, doi:10.1029/2010JD014439, 2010.

**More minor comments:**

9. A suggestion for consideration, since from my understanding, ACP does not limit the abstract length: the abstract as it stands right now reads more like a conclusion/summary. Could it be pared down more? (for instance, what are the three key findings of this study?) Reply: Thank you for the suggestion. We revise the abstract by summarizing and highlighting main findings from this study.

10. The title states "...an analysis combining observations with regional modeling," but as I understand it, the observations were not used to bias correct or calibrate the model in any way. In this sense the observations were used purely for model validation, and as such this wording is perhaps misleading (there's no "combining" involved). Instead, perhaps something more along the lines of "model validation and regional modeling" would be more accurate, if the authors decide to retain a focus on the model validation part.

Reply: Yes, the observations are mainly for model validation and interpretation, we revise the title to "Seasonal characteristics of emission, distribution and radiative effect of marine organic aerosols over the western Pacific Ocean: an investigation with a coupled regional climate-aerosol model".

11. L145-147: "supposed to" is not a good word choice here. If one wants to express uncertainty, "may" could be a good replacement. Reference(s) for this claim should also be added.

Reply: revised.

12. L166-168: the internally mixed anthropogenic aerosol is assumed to have a fixed distribution that does not change shape following activation and sedimentation? If yes, has there been studies justifying this choice? I wonder if the anthropogenic aerosol may be represented by size bins, as is done for the natural aerosols. If yes, please clarify.

Reply: In the current version of RIEMS-Chem, a bulk method is used to represent internally mixed anthropogenic aerosols, with a typical geometric diameter (standard deviation) fitted by a lognormal distribution based on recent observations in east China (Ma et al., 2017; Wu et al., 2017), which found that the geometric mean radius of a dry aerosol internal mixture during the periods from light/moderate to severe pollution stages increased slightly from 0.10 to 0.12  $\mu$ m, so a geometric mean radius of 0.11  $\mu$ m with a standard deviation of 1.65 are chosen for the internal mixture of anthropogenic aerosols, as that in Li et al. (2020).

**References**

- Ma, Q. X., Wu, Y. F., Zhang, D. Z., Wang, X. J., Xia, Y. J., Liu, X. Y., Tian, P., Han, Z. W., Xia, X. A., Wang, Y., and Zhang, R. J.: Roles of regional transport and heterogeneous reactions in the PM2.5 increase during winter haze episodes in Beijing, Sci. Total. Environ., 599/600, 246–253, 2017.
- Wu, Y. F., Wang, X. J., Tao, J., Huang, R. J., Tian, P., Cao, J. J., Zhang, L. M., Ho, K. F., Han, Z. W., Zhang, R. J.: Size distribution and source of black carbon aerosol in urban Beijing during winter haze episodes, Atmos. Chem. Phys., 17, 7965-7975, 2017.
- Li Jiawei, Han Zhiwei, Wu Yunfei, Xiong Zhe, Xia Xiangao, Li Jie, Liang Lin, Zhang Renjian: Aerosol radiative effects and feedbacks on boundary layer meteorology and PM2.5 chemical components during winter haze events over the Beijing-Tianjin-Hebei region. Atmos. Chem. Phys., 20, 8659–8690, 2020.

13. L168: "Natural aerosols (mineral dust and sea salt)" and MPOA? Also, in general, how do MPOA and MSOA fit in these model descriptions? E.g. on L175: what's the hygroscopicity of MPOA and MSOA? The same as POA and SOA?

Reply: We revise the sentence to "Mineral dust and sea salt are represented by .....". The hygroscopicity for MPOA (0.1) and MSOA (0.2) are assumed to be the same as those for anthropogenic POA and SOA, we add relevant information in section 2.2 in the revised version.

14. L210: OMss is the organic mass fraction of sea spray aerosol, not sea salt. Reply: Yes, revised.

15. L220-221: please add reference for the OM/OC ratio Reply: Added.

**Reference**

Gantt, B. and Meskhidze, N.: The physical and chemical characteristics of marine primary organic aerosol: a review, Atmos. Chem. Phys., 13, 3979–3996, 2013.

16. L262: please add reference/justification of choice for the MSOA soluble mass fraction

Reply: So far, our knowledge on the properties of marine organic aerosols is very limited, especially in the western Pacific Ocean. Facchini et al. (2008) found that the OC within submicron particles generated by bubble bursting is mainly water insoluble (on average 94  $\pm$  4% of total carbon) based on bubble bursting experiments during a phytoplankton bloom in the North Atlantic, which also denotes approximately 10% of OC is soluble.

In this study, we assume that MPOA is slightly soluble with a solubility of 0.1 considering aging processes it may undergo (Gantt and Meskhidze, 2013). Because SOA is more hygroscopic than POA, with respect to that the hygroscopicity ( $\kappa$ ) of SOA is about twice that of POA (Liu and Wang, 2010; Westervelt et al., 2012), we assume the solubility of MSOA is twice that of MPOA. The uncertainty in chemical properties of MOA and its potential effect on IRE is discussed above and in the response to the second reviewer.

References

- Facchini, M. C., Rinaldi, M., Decesari, S., Carbone, C., Finessi, E., Mircea, M., Fuzzi, S., Ceburnis, D., Flanagan, R., Nilsson, D., de Leeuw, G., Martino, M., Woeltjen, J., and O'Dowd, C. D.: Primary sub-micron marine aerosol dominated by insoluble organic colloids and aggregates, Geophys. Res. Lett., 35, L17814, doi:10.1029/2008GL034210, 2008.
- Liu X. and Wang J.: How important is organic aerosol hygroscopicity to aerosol indirect forcing, Environ. Res. Lett. 5, 044010, doi:10.1088/1748-9326/5/4/044010, 2010.
- Westervelt, D. M., Moore, R. H., Nenes, A., and Adams, P. J.: Effect of primary organic sea spray emissions on cloud condensation nuclei concentrations, Atmos. Chem. Phys., 12, 89–101, 2012.

17. L282-283: perhaps I'm missing something, but if aerosol activation/Nc is already calculated based on the Abdul-Razzak and Ghan scheme, why is "the number of aerosols activated assumed to be equal to the number of aerosols scavenged in cloud"?

Reply: We are sorry for the confusion. What we want to express is 'the activated aerosols (into cloud droplet) are removed from the air, the number of removed airborne aerosol is equal to that activated", we revised the relevant sentence.

18. L359: Pearson correlation coefficient? Please be specific. Reply: Yes.

19. L718-719 and Figure 7: how does the sea salt emission look like? Perhaps add a column in figure 7 for sea salt emission (since EM\_POA=  $\alpha \times Ess \times OMss$ )?

Reply: Thank you, we add a column in Figure 7 for sea salt emission following your suggestion in the revised version.

20. L756-759: please clarify if this is a speculation or confirmed by analysis. It's difficult to tell by the wording.

Reply: We check the data and find that in the NWP region, the Chl-a concentration is negatively correlated with sea salt emission in MAM, whereas they are positively correlated

in SON. This could lead to the slightly higher MPOA emission (related to both Chl-a concentration and sea salt emission) in SON than that in MAM. We have revised the relevant description in the manuscript.

21. L777: "annual mean" as in "ng m-2 s-1" averaged over the area and over the whole year? Or "annual emission" in "Tg y-1"? Reply: It is "annual emission in Tg y-1".

22. L788-790: what is the relevance of this sentence in the context of the current study? Does this then imply that it is reasonable to compare simulation to observation from a different year for order of magnitude check?

Reply: This sentence is not relevant to the model results, so we delete it in the revised version.

23. Figures 4 and 5: do the standard deviations represent variability of the monthly/seasonal mean across multiple years or also the variability within each month/season?

Reply: We draw the standard deviations in Figures 4 and 5 based on raw data presented in the publication, these data were sampled on a bi-weekly basis in the islands, so the deviation could represent bi-weekly variability.

24. Figure 9: "mean monthly" (cm month-1; total monthly precipitation averaged over multiple months) instead of "monthly mean" (mm h-1; average precipitation rate over each month). Note both in figure title and caption.

Reply: Yes, it is an average over three months in each season, we revise it both in figure title and caption in the revised version.

25. Table 9: "cm grid-1 month-1": do the authors mean cm month-1?

Reply: It means "the sum of gridded monthly total precipitation divided by the total grid number in a specific region, so it represents cm month-1 for a certain region". The unit is revised.

**Response to Referee #2**

**General comments:**

This study focuses on primary and secondary marine aerosols over the Western Pacific Ocean and their radiative effect. The findings are interesting and this study is well suited for ACP. In its present form, I found this study is too lengthy which makes it challenging to read. Important methodological information and analysis are missing regarding the radiative impact of MOA.

Reply: Thank you for your pertinent and valuable comments which help us improve the manuscript. We address your comments carefully and revise the manuscript accordingly by taking your suggestions into account.

Abstract is very long. I recommend the authors try to highlight no more than 2-3 key points. **Reply:** Thank you. We revise the abstract by highlighting several key points.

line 25. It reads like a new finding but as far as I understand this just describes the parameterization of Gantt et al. This should be clarified

Reply: Sorry for the description, we delete this sentence in the revised version.

line 70. The introduction focuses primarily on literature published prior to 2012. A lot has been done both in terms of observations (field and lab) and in terms of parameterization since that needs to be discussed by the authors.

Here are a couple of studies (by no mean an exhaustive list) that the authors may want to consider

Conte, L., Szopa, S., Aumont, O., Gros, V., & Bopp, L. (2020). Sources and sinks of isoprene in the global open ocean: Simulated patterns and emissions to the atmosphere. Journal of Geophysical Research: Oceans, 125, e2019JC015946. https://doi.org/10.1029/2019JC015946 Bates, T. S., Quinn, P. K., Coffman, D. J., Johnson, J. E., Upchurch, L., Saliba, G., et al. (2020). Variability in Marine Plankton Ecosystems Are Not Observed in Freshly Emitted Sea Spray Aerosol Over the North Atlantic Ocean. Geophysical Research Letters, 47, e2019GL085938. https://doi.org/10.1029/2019GL085938

Quinn, P. K., Bates, T. S., Coffman, D. J., Upchurch, L., Johnson, J. E., Moore, R., et al. (2019). Seasonal variations in western North Atlantic remote marine aerosol properties. Journal of Geophysical Research: Atmospheres, 2019; 124: 14240–14261. https://doi.org/10.1029/2019JD031740.

Brüggemann, M., Hayeck, N. & George, C. Interfacial photochemistry at the ocean surface is a global source of organic vapors and aerosols. Nat Commun 9, 2101 (2018). https://doi.org/10.1038/s41467-018-04528-7

Betram et al. Sea spray aerosol chemical composition: elemental and molecular mimics for laboratory studies of heterogeneous and multiphase reactions. Chemical Society Reviews, 31 Mar 2018, 47(7):2374-2400 DOI: 10.1039/c7cs00008a

Quinn, P., Coffman, D., Johnson, J. et al. Small fraction of marine cloud condensation nuclei made up of sea spray aerosol. Nature Geosci 10, 674–679 (2017). https://doi.org/10.1038/ngeo3003

S. M. Burrows and O. Ogunro and A. A. Frossard and L. M. Russell and P. J. Rasch and S. M. Elliott A physically based framework for modeling the organic fractionation of sea spray aerosol from bubble film Langmuir equilibria Atmospheric Chemistry and Physics 14, 13601–13629 (2014). https://doi.org/10.5194/acp-14-13601-2014

**Reply:** We are sorry for missing these papers, and thank you very much for telling us recent progresses on this issue in both observation and parameterization. They are valuable and helpful for this study.

We revise the introduction in the revised version by including these references in the revised version, which is described as "However, Quinn et al. (2014) found that the organic carbon content of sea spray aerosol is weakly correlated with satellite retrieved chlorophyll-a concentration based on cruise measurements in the North Atlantic Ocean and the coastal waters of California. Bates et al (2020) reported that plankton bloom has little effect on the emission flux, organic fraction or cloud condensation nuclei of sea spray aerosol based on cruise experiment over the North Atlantic. Burrows et al. (2014) developed a novel physically

based framework for parameterizing the organic fractionation of sea spray aerosol by consideration of ocean biogeochemistry processes, and their predicted relationships between Chl-a and organic fraction are similar to existing empirical parameterizations associated with ocean Chl-a concentrations at high Chl-a levels, but the empirical relationships may not be adequate to predict OM fraction of sea spray aerosol outside of strong seasonal blooms. Considering the strong bloom seasonality in the western Pacific region and the availability of global satellite data for Chl-a concentration, and the lack of cruise measurements on the relationship between sea spray organic aerosol fluxes and Chl-a in this region, we adopted the scheme of Gantt et al (2011) for parameterizing marine primary organic aerosol emission in this study."

We also add publications on CCN activity of sea spray aerosol as "Based on the measurements from seven research cruises over the Pacific, Southern, Arctic and Atlantic oceans between 1993 and 2015, Quinn et al. (2017) indicated that sea spray aerosol generally makes a contribution of less than 30% to CCN population at supersaturation of 0.1 to 1.0% on a global basis"

The revised introduction includes more relevant studies on this issue, although their findings or conclusions may be different.

line 169 It would be worth mentioning that this range exceeds the valid range for Monahan (0.8 < r80 < 10 mm)

Reply: We are sorry for missing the relevant information. We actually apply the scheme of Gong (2003) in this study, which was an improvement of the Monahan et al. (1986) scheme. The valid range in the Gong (2003) scheme is from  $0.07\mu m$  to  $20\mu m$  radius at RH = 80%. We add the description and relevant reference in the revision.

**Reference**

Gong, S.L., 2003. A parameterization of sea-salt aerosol source function for sub- and super-micron particles, Global Biogeochem. Cy., 17(4), 1097. doi:10.1029/2003GB002079.

line 202 GEIA is a portal for many different inventories. Are the authors using a climatology of MEGANv2? That would be surprising since very detailed year-specific inventories are used for anthropogenic and biomass burning emissions. Please clarify.

Reply: Sorry for the confusion. The biogenic VOC emission is derived from the CAMS-BIO Global biogenic emissions dataset (CAMS-GLOB-BIO v3.1) (Granier et al., 2019; Sindelarova et al., 2014) distributed by ECCAD-GEIA (https://permalink.aeris-data.fr/CAMS-GLOB-BIO, last access: 2020/02/10) and the monthly mean biogenic emission for the year 2014 with a horizontal resolution of  $0.25^{\circ}$  is used, which is consistent with other year-specific inventories. We revise relevant description and add the references in the revision.

**References**

Granier, C., S. Darras, H. Denier van der Gon, J. Doubalova, N. Elguindi, B. Galle, M. Gauss, M. Guevara, J.-P. Jalkanen, J. Kuenen, C. Liousse, B. Quack, D. Simpson, K. Sindelarova, 2019. The Copernicus Atmosphere Monitoring Service global and regional emissions (April

2019 version), Report April 2019 version, doi:10.24380/d0bn-kx16.

Sindelarova, K., Granier, C., Bouarar, I., Guenther, A., Tilmes, S., Stavrakou, T., Müller, J.-F., Kuhn, U., Stefani, P., and Knorr, W., 2014. Global data set of biogenic VOC emissions calculated by the MEGAN model over the last 30 years. Atmos. Chem. Phys., 14, 9317–9341. https://doi.org/10.5194/acp-14-9317-2014.

line 226 It seems there could be a lot of other reasons for this difference. MODIS vs VIIRS Chl-A, differences in wind speed and sea salt parameterizations.

Reply: Yes, we delete "The large difference in the choice of  $\alpha$  suggests that the emission rate of MPOA could be very regionally dependent" and move the description of the scheme to the support material in the revised version.

line 229 This section completely ignores the abiotic source of isoprene (see references above), which may be as large if not larger than the biological source in marine environments.

Reply: Thank you. We only consider marine isoprene from biological source in this study. The marine abiotic source of isoprene (due to photochemical production in the sea surface microlayer) may be important according to recent studies (Brüggemann et al. 2018; Conte et al., 2020), however, the production mechanism is still highly uncertain. Brüggemann et al. (2018) estimated a global total oceanic isoprene emission of 1.11 Tg yr-1 from both biological and photochemical production. It appears that the scheme of Gantt et al. (2009) predicted a higher biological isoprene emission, because their estimate (0.92 Tg yr-1) is close to that in Brüggemann et al. (2018). We add a description on the abiotic source of isoprene with relevant publications in section 2.3 in the revised version as "The marine abiotic source of isoprene (due to photochemical production in the sea surface microlayer) may be important according to recent studies (Brüggemann et al., 2018; Conte et al., 2020), which is not considered in this study because the production mechanism for marine abiotic isoprene is poorly understood at present."

line 265 I may be missing something but I am not sure how to reconcile the source function for sea salt (radius > 0.1um) which is used to derive MOA emissions, with the assumed diameter of MOA (0.1 um). Please clarify.

Reply: we are sorry for missing such information. As we explained above, we use the scheme of Gong (2003) developed based on Monahan et al. (1986), so the sea salt radius range is from  $0.07\mu m$  to  $20\mu m$ . Only fine model MOA (~ $0.1\mu m$ ) is considered in this study.

line 267 do the authors also use 5 size bins to represent MPOA or do they only consider sub-micron MPOA for this work?

Reply: We only consider the sub-micron MPOA because the cruise measurements over the marginal seas of China and the western Pacific (Feng et al., 2017) revealed that TOC mass (mainly in MPOA) is mainly concentrated in the sub-micron size and the super-micron TOC was generally below the detection limit.

Section 3. I suggest to have a section devoted solely to describing the different sources of observations, such that section 3 can focus solely on the model performance. The analysis of

BC should focus more clearly on how these observations can help understand marine organic aerosols. The detailed BC analysis presented here could be moved to supporting materials, which would help shorten the manuscript.

Reply: Thank you for the good suggestion. We add section 2.5 for describing the different sources of observations, and shorten the manuscript by moving Figure 4 (in previous version) to the supplement and by deleting relevant description on model validation for BC.

It would be helpful to evaluate the simulated Na+ with the UMiami/Prospero dataset.

Reply: Yes, following your suggestion, we collected bi-weekly Na+ measurements at Japan islands from the Acid Deposition Monitoring Network in East Asia (EANET) and compared with our model simulations. We add a brief description on this comparison in section 3.1 in the revision as "Comparison with observations of Sodium (Na+) concentration at 6 Japan coastal/island sites from EANET is conducted to further examine the model performance for sea salt. The modeled sodium is estimated to be 38.56% of sea salt mass (Kelly et al., 2010), and the agreement between observation and model simulation is generally satisfactory at all sites except at Oki in December, when the model largely underpredict Na+. The model well reproduces the seasonality of sodium concentration, with the maximum in winter and the minimum in summer (Figure 3). The model predicts sodium concentration best at Ogasawara, with the correlation coefficient of 0.85 and NMB of 5%. The overall correlation coefficient for all sites is 0.50, with NMB of -11% (Table S1)". Figure 3 (below) and a table for statistics in the supplement (Table S1) are also added in the revised version.